# Altering microtubule dynamics is synergistically toxic with spindle assembly checkpoint inhibition

Klaske M Schukken[1], Yu-Chih Lin[2,*], Petra L Bakker[1,*], Michael Schubert[1], Stephanie F Preuss[1], Judith E Simon[1], Hilda van den Bos[1], Zuzana Storchova[3], Maria Colomé-Tatché[1,4,5], Holger Bastians[2], Diana CJ Spierings[1], Floris Foijer[1]

**Chromosomal instability (CIN) and aneuploidy are hallmarks of cancer. As most cancers are aneuploid, targeting aneuploidy or CIN may be an effective way to target a broad spectrum of cancers. Here, we perform two small molecule compound screens to identify drugs that selectively target cells that are aneuploid or exhibit a CIN phenotype. We find that aneuploid cells are much more sensitive to the energy metabolism regulating drug ZLN005 than their euploid counterparts. Furthermore, cells with an ongoing CIN phenotype, induced by spindle assembly checkpoint (SAC) alleviation, are significantly more sensitive to the Src kinase inhibitor SKI606. We show that inhibiting Src kinase increases microtubule polymerization rates and, more generally, that deregulating microtubule polymerization rates is particularly toxic to cells with a defective SAC. Our findings, therefore, suggest that tumors with a dysfunctional SAC are particularly sensitive to microtubule poisons and, vice versa, that compounds alleviating the SAC provide a powerful means to treat tumors with deregulated microtubule dynamics.**

## Introduction

Chromosomal INstability (CIN) is the process through which chromosomes mis-segregate during mitosis. CIN leads to cells with an abnormal DNA content, a state known as aneuploidy. As three of four cancers are aneuploid (Weaver & Cleveland, 2006; Foijer et al, 2008; Duijf et al, 2013), CIN is considered an important contributor to tumorigenesis. Indeed, CIN has been associated with metastasis (Bloomfield & Duesberg, 2016; Xu et al, 2016), increased probability of drug resistance (Lee et al, 2011; Sansregret & Swanton, 2017) and generally, a lowered patient survival (Carter et al, 2006; Walther et al, 2008; McGranahan et al, 2012). While the frequent occurrence

of CIN and resulting aneuploidy in cancer is generally attributed to the acquired ability of cancer cells to adapt their palette of oncogenic features as the tumor evolves, ongoing chromosome mis-segregation also has negative effects on cancer cells. The downside of CIN for cancer cells is that most newly acquired karyotypes lead to reduced proliferation (Torres et al, 2007; Williams et al, 2008; Foijer et al, 2017) and induction of aneuploidy-imposed stresses (Torres et al, 2010). In addition to this, ongoing missegregation causes further structural DNA damage (Zhang et al, 2015; MacKenzie et al, 2017) that, together with unfavorable karyotypes, leads to cell death (Kops et al, 2004; Burds et al, 2005; Santaguida et al, 2017) or senescence (Andriani et al, 2016).

To protect from CIN, cells have mechanisms in place that maintain proper chromosome inheritance. The Spindle Assembly Checkpoint (SAC) is one such mechanism preventing CIN by inhibiting the onset of anaphase until all chromosomes are properly attached to the two opposing spindle poles, reviewed in detail by Musacchio and Salmon (2007). Interfering with the SAC, for instance, by inactivating key components of the checkpoint leads to frequent chromosome mis-segregation events and is commonly used to study the consequences of CIN in vitro and in vivo (Kops et al, 2004; Foijer et al, 2013, 2014, 2017).

Although SAC impairment is rare in human cancer (Gordon et al, 2012), many cancers show signs of a partially impaired SAC, for instance, as a result of increased expression of proteins with a direct role in the SAC or their regulators, such as Rb mutations that lead to increased expression of Mad2, and thus, provoke a CIN phenotype (Pfau & Amon, 2012). Furthermore, altered microtubule (MT) dynamics are another source of CIN in many cancers (Bakhoum et al, 2009; Ertych et al, 2014; Stolz et al, 2015) as restoring tubulin dynamics to normal levels can decrease CIN rates in many cancer cell lines (Ertych et al, 2014). Conversely, commonly used cancer drugs such as paclitaxel or vincristine interfere with MT polymerization rates, thus increasing CIN rates in cancer cells. This observation suggests

[1]European Research Institute for the Biology of Ageing, University of Groningen, University Medical Centre Groningen, Groningen, The Netherlands    [2]Goettingen Center for Molecular Biosciences and University Medical Center, Goettingen, Germany    [3]Department of Molecular Genetics, University of Kaiserslautern, Germany    [4]Institute of Computational Biology, Helmholtz Center Munich, German Research Center for Environmental Health, Neuherberg, Germany    [5]Technical University of Munich, School of Life Sciences Weihenstephan, Technical University of Munich, Freising, Germany

Correspondence: f.foijer@umcg.nl
*Yu-Chih Lin and Petra L Bakker contributed equally to this work

that imposing CIN phenotypes onto cancer cells is a powerful strategy to eradicate tumors. However, it is not yet clear whether exacerbating CIN in cells with a preexisting CIN phenotype is wise or not.

As CIN and aneuploidy discriminate cancer cells from healthy cells, both make for attractive targets for cancer therapy. To reveal potential general vulnerabilities of aneuploid cells, Tang et al (2011) performed a small molecule compound screen, which revealed the energy stress-inducing compound 5-Aminoimidazole-4-carboxamide ribonucleotide (AICAR) to be more toxic to aneuploid cells than euploid cells (Tang et al, 2011). This aneuploidy-specific toxicity was shown to be true in cell culture experiments as well as in cancer mouse models, a promising result for future aneuploid cancer therapies.

Although CIN and aneuploidy are intimately related, CIN has additional effects on cell physiology and growth in addition to those imposed by the resulting aneuploidy (Schukken and Foijer, 2018). Because CIN drives karyotype heterogeneity, thus increasing the rate of evolution that cancer cells use to acquire new features and adapt (McGranahan et al, 2012; Giam & Rancati, 2015), targeting CIN would provide an even more powerful means to kill cancer cells than aneuploidy alone.

In this study, we, therefore, performed two small-scale drug screens, one to identify small molecule compounds that target aneuploid cells and another to find compounds that are more toxic to CIN cells than to chromosomally stable cells. For this purpose, we selected a collection of drug-like molecules from a list of drugs already being used in the clinic or in advanced-stage clinical trials. Compounds were further selected for their potential role in targeting CIN or aneuploid cells, such as targeting cell survival (Dekanty et al, 2012; Foijer et al, 2013), proliferation (Williams et al, 2008; Ben-david et al, 2014; Gogendeau et al, 2015; Sheltzer et al, 2017), protein processing (Oromendia et al, 2012; Stingele et al, 2012), DNA repair (Bakhoum et al, 2014, 2018), transcriptional deregulation (Upender et al, 2004; Stingele et al, 2012), and cellular metabolism (Williams et al, 2008; Tang et al, 2011) as these processes are typically deregulated in aneuploid cells. Indeed, our screen for aneuploidy-targeting compounds revealed a compound targeting cellular metabolism, validating earlier findings from the Amon laboratory (Tang et al, 2011). Furthermore, the CIN screen revealed that the Src inhibitor SKI606 (bosutinib) is synergistically toxic to cells with an alleviated SAC. We find that the mechanism underlying the toxicity of SKI606 in SAC-deficient cells results from deregulated tubulin polymerization rates imposed by Src inhibition. Our results, therefore, indicate that combining SAC inhibition with tubulin deregulation is synergistically toxic to cells and might provide a powerful means to target cancer cells with a CIN phenotype.

# Results

CIN and the resulting aneuploidy lead to a deregulated transcriptome and proteome (Tang et al, 2011; Stingele et al, 2012; Foijer et al, 2013, 2014) and can provoke cell cycle delay, senescence, or apoptosis (Giam & Rancati, 2015; Andriani et al, 2016; Santaguida et al, 2017; Chunduri & Storchová, 2019). Furthermore, ongoing CIN can lead to further DNA damage (Zhang et al, 2015; MacKenzie et al, 2017). We, therefore, reasoned that targeting RNA or protein processing, transcriptional regulation, apoptosis, or DNA repair might be particularly toxic to aneuploid cells and cells exhibiting a CIN phenotype. As CIN and aneuploidy are different concepts (Schukken & Foijer, 2018) and have different consequences for cells (Stingele et al, 2012; Andriani et al, 2016; Schukken & Foijer, 2018) aneuploidy and CIN might impose different therapeutic vulnerabilities. To test this, we performed two small-scale drug screens, one to identify compounds that selectively prevent the propagation of aneuploid cells and another to identify small molecules that selectively targets CIN cells.

## A small-scale drug screen to identify compounds that selectively target aneuploid cells

We first selected 95 drug-like molecules from a drug library composed of drugs that target processes that aneuploid or CIN cells might rely on and are already being used in the clinic or being tested in clinical trials (Table S1). Next, we determined the initial drug concentration for each drug to be used in the screen. For this, we exposed wild-type RPE1 cells (a near-diploid non-cancer cell line derived from retinal epithelium [Soto et al, 2017]) to decreasing concentrations of the drugs, starting at 10 $\mu M$ for all compounds, and compared cell proliferation of drug-exposed cells to proliferation of DMSO-treated cells over a period of 7 d. We purposely chose an untransformed cell line, as this allows studying the combinational effect of CIN and drugs in an otherwise unperturbed setting.

Next, we subjected stable aneuploid RPE-1 cells, trisomic for chromosomes (chrs.) 5 and 12 (Fig S1A, [Stingele et al, 2012]), to the same drug-treatment regime and compared proliferation between diploid and aneuploid RPE1 cells (Supplemental Data 1) using an IncuCyte high content imager. Fig S1B schematically shows the experimental design and analysis approach. Note that aneuploid RPE1 cells showed a modestly reduced proliferation rate compared with control RPE1 cells (Fig S1C) in line with earlier observations (Williams et al, 2008) for which we corrected when analyzing the growth curves. To quantify differences between wild-type and aneuploid RPE1 cells, we compared the area under the curve (AUC) as a measure of cumulative cell growth (Fig 1A and B) and the slope of the logarithmic growth as a measure for the proliferation rate (Fig 1C and D), also see the Materials and Methods section. While this screen revealed some drugs for which aneuploid RPE1 cells were more sensitive ($\log_2 > 0$; $P < 0.05$) or less sensitive ($\log_2 < 0$; $P < 0.05$), we only found one compound (#2379, ZLN005; a transcriptional regulator of PGC-1$\alpha$) for which the effect was significant after Bonferroni multiple testing correction (Fig 1A) in one of the two screens. The combined effects of aneuploidy and ZLN005 act synergistically as assessed by a Bliss independence test (50% stronger effect than additive, $P$ = 3.2E-3, [Zhao et al, 2014]). Indeed, further validation confirmed the selective growth defect of aneuploid RPE1 cells imposed by ZLN005 (Fig 1E). However, as ZLN005 targets energy metabolism, very similar to what others have found for AICAR (Tang et al, 2011), we did not pursue this compound further. We, therefore, conclude that our aneuploidy screen did not uncover novel targetable vulnerabilities of aneuploid cells and next performed a screen for compounds that selectively inhibits CIN cells.

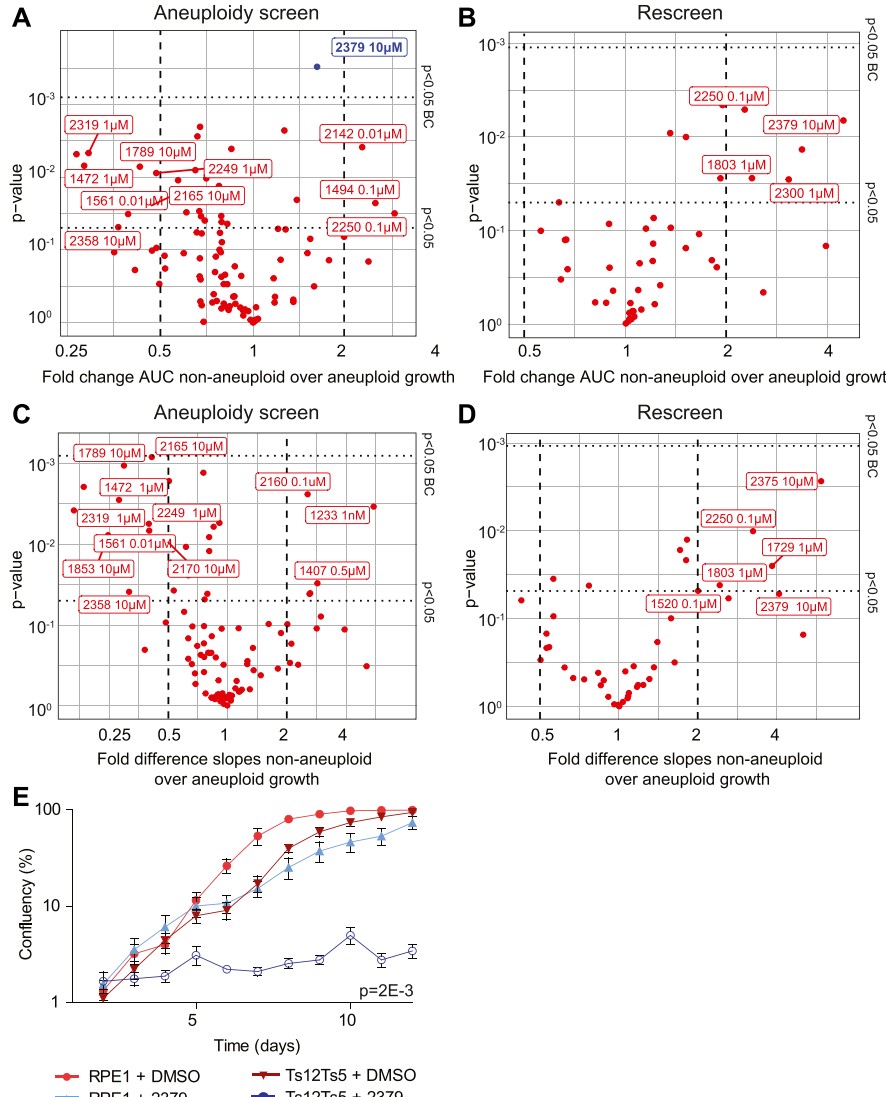

**Figure 1.   Aneuploid cells are sensitive to a metabolism-enhancing drug.**
**(A, B, C, D)** RPE1 control cells and stable aneuploid RPE1 Ts12 Ts5 cells were screened with 95 drugs, each drug screened in triplicate. 45 drugs were rescreened. The *P*-values and the effect size for drug's effect on RPE1 and RPE1 Ts12 Ts5 cells were plotted. **(A, B, C, D)** Data were analyzed through quantification of area under the curve (AUC, A, B), and slope analysis (C, D) of both the initial screen (A, C) and rescreened drugs (B, D). Drugs with difference >1 and *P*-value < 0.05 after Bonferroni correction are indicated in blue. **(E)** Validation growth curves of RPE1 control and RPE1 Ts12 Ts5 cells with and without 10 *μM* 2,379. Data were obtained by sequential daily microscope images and analyzed by FIJI-PHANTAST. All data involve at least three biological replicates, each with three technical replicates. Error bars indicate SEM. *P*-values are calculated in two-sided *t* test for AUC, correcting for cell line control. DMSO control curves are shared with Figs S1C and S3G.

## A conditional Mad2 knockdown cell line to model CIN

To screen for compounds that selectively target cells with a CIN phenotype, we needed a cell line in which CIN can be provoked in an inducible fashion, as long-term CIN phenotypes are typically selected against in tissue culture (Kops et al, 2004; Foijer et al, 2014). For this, we engineered RPE1 hTert cells in which the SAC can be inhibited through expression of a Doxycycline-inducible Mad2 shRNA construct, from here on referred to as Mad2 conditional knockdown (Mad2cKD) RPE1 cells. Mad2 knockdown efficiency was quantified by quantitative PCR (Fig 2A) and Western blot (Fig 2B), which revealed that Mad2 levels were reduced by 90% within 3 d of doxycycline treatment. To test whether Mad2 inhibition was sufficient to alleviate the SAC, we exposed cells to the MT poison nocodazole, determined accumulation in mitosis by quantifying phospho-histone H3 using flow cytometry, and found that dox-treatment for 3 d or longer was sufficient to completely alleviate the SAC in Mad2cKD RPE1 cells (Fig 2C). Therefore, for all follow-up

experiments involving Mad2cKD RPE1 cells, the cells were pre-treated with doxycycline for a minimum of 3 d before drug administration. As expected, we found that Mad2cKD moderately decreased cell proliferation (~25%), which we corrected for in our downstream analyses in each of our experiments (Fig S2A). Next, we determined whether SAC inhibition in Mad2cKD RPE1 cells indeed leads to a CIN phenotype. To this aim, we quantified interphase and mitotic abnormalities using live cell imaging (Fig 2D and E). Indeed, Mad2cKD cells displayed a significantly increased CIN-rate: 46% of the Mad2cKD RPE1 cells displayed mitotic abnormalities compared with only 1% of control cells. In addition, the fraction of cells with interphase remnants of mitotic aberrations such as micronuclei increased from 2 to 24%. Finally, we quantified aneuploidy by single-cell whole genome sequencing (Bakker et al, 2016; van den Bos et al, 2016). Whereas control RPE1 cells show little aneuploidy (2 of 114 cells sequenced) except for a known structural abnormality for chromosome 10 (Fig 2F and Worrall et al (2018)), 45% of dox-treated Mad2cKD cells displayed one to few aneusomies per cell (76 of 169

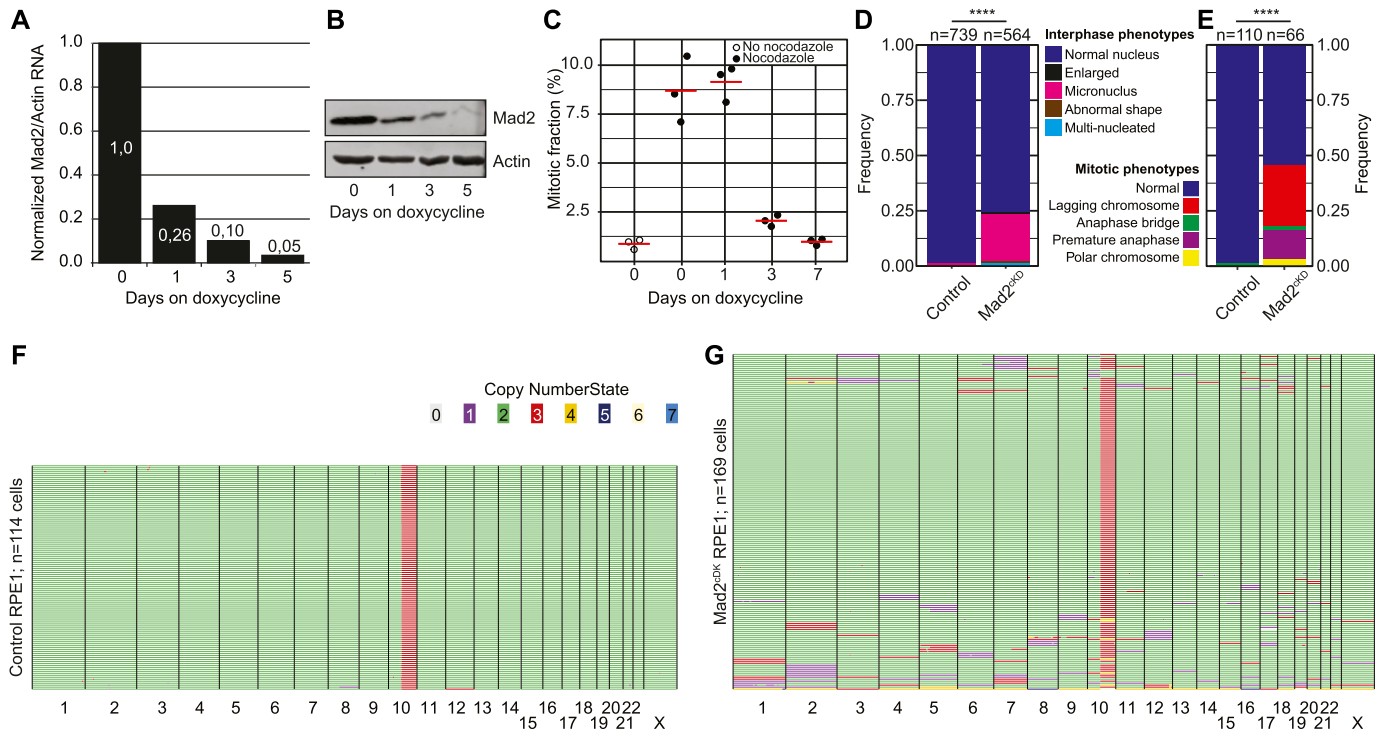

**Figure 2. Engineering a cell line for conditional CIN.**
**(A)** Quantitative PCR for Mad2 RNA levels over time in Mad2^cKD RPE1 cells. **(B)** Western blot for Mad2 levels over time in RPE1 in Mad2^cKD RPE1 cells. **(C)** Mitotic accumulation of nocodazole-challenged control and Mad2^cKD RPE1 cells measured by phosphorylated histone H3. **(D, E)** Quantification of mitotic phenotypes of control and Mad2^cKD RPE1 cells assessed by time-lapse imaging for interphase cells (D) and mitotic cells (E). "n" refers to the number of cells analyzed, *P*-values from chi-squared test. Data also displayed in Fig 5H. **(F, G)** Single-cell whole-genome sequencing data quantified by AneuFinder for RPE1 control cells (F, 114 cells, 2 aneuploid) and Mad2^cKD RPE1 cells after 5 d of doxycycline treatment (G, 169 cells, 76 aneuploid). Colors refer to the copy number state for each chromosome (fragment).

cells, Fig 2G) within 5 d after induction of the Mad2 shRNA, confirming a substantial CIN phenotype. Together, these features make the Mad2^cKD cells highly suitable to screen for compounds that target CIN cells.

### The Src inhibitor SKI606 selectively targets Mad2^cKD cells

We next used the Mad2^cKD RPE1 cells to screen for compounds that selectively inhibit the expansion of CIN cells (Fig S2B). For this, we exposed control and Mad2^cKD RPE1 cells to 58 compounds (Table S2) and compared the maximum proliferation rate and cumulative cell number between Mad2^cKD RPE1 cells and control RPE1 cells using the same setup as for the aneuploidy screen described above. To assess both short-term and longer term effects of the drugs, we quantified proliferation and cumulative cell number over the first 4 d and over days 5–8 separately (Fig S2B). Intriguingly, we found that the mTor inhibitor AZD8055 (compound #1561) at 0.1 μM acted synergistically with CIN in reducing cell numbers (31% greater than additive effect; *P* = 2.7E-4, Bliss independence test) during the first 4 d of the screen, but became fully toxic to both control and Mad2^cKD cells from day 5 onward (Fig 3A and B and Supplemental Data 2 for all growth curves). Conversely, we found that the Src inhibitor SKI606 (compound #1407) at 0.1 μM acted synergistically with CIN (48% greater effect than additive; *P* = 7.3E-3, Bliss independence test) during the second half of the screen (Fig 3C and D and Supplemental Data 2) and less so during the first half of the screen.

Note that the observed effects were not related to the doxycycline treatment required to induce Mad2 shRNA, as doxycycline alone had no effect on proliferation (Fig S3A). Next, we wanted to validate our findings in independent growth assays. In addition to AZD8055 and SKI606, we also retested compounds #2180 (TMP195, HDAC inhibitor), #2250 (CHR6494 trifluoroacetate, Haspin inhibitor), #2831 (EPZ015666, Prmt5 inhibitor), #1801 (pyroxamide, HDAC 1 inhibitor), #1803 (MS 275, HDAC 1 and 3 inhibitor), and #2008 (Tenovin 1, SIRT 1 & 2 inhibitor) that also showed some effect in the primary CIN screen. For these validation experiments, proliferation was quantified by daily cell confluency measurements from microscope images as described in the Materials and Methods section. These experiments revealed that although SKI606 (#1407), AZD8055 (#1561), and EPZ015666 (#2831) reproducibly inhibited the growth of Mad2^cKD RPE1 cells more than that of control cells (Fig 4A–F), this was not the case for TMP195, CHR6494, pytoxamide, MS 275, and Tenovin (Fig S3B–G). Given that SKI606 gave the largest growth inhibitory effect on Mad2^cKD RPE1 cells, most notably at 0.5 μM (Fig 4E), we decided to further pursue this compound. It is interesting to note that SKI606 had no significant effect on stable aneuploid cells (Fig S3G–J), and vice versa, that ZLN005 (#2379), identified in the aneuploidy screen, had no significant effect on Mad2^cKD CIN cells (Fig S3K), suggesting that compounds that are selectively toxic to stable aneuploid cells not necessarily target CIN cells, and vice versa.

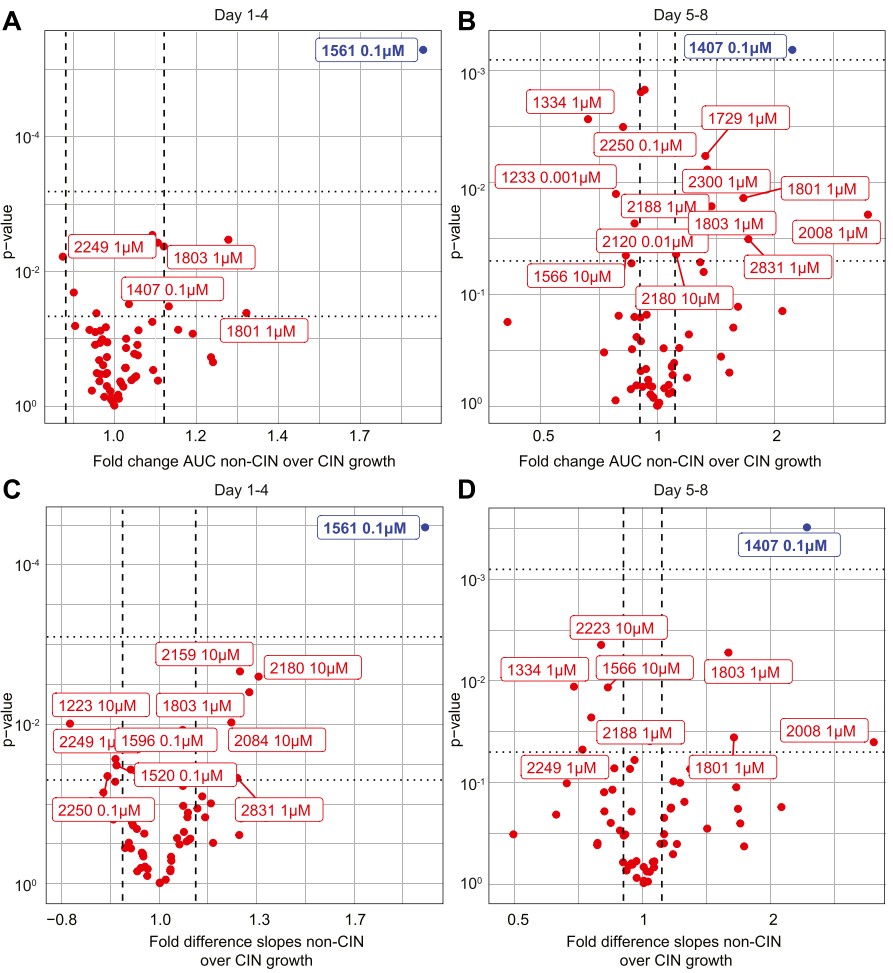

**Figure 3. Screen for compounds that selectively kill CIN cells reveals several candidates.**
**(A, B, C, D)** Growth curves of control and Mad2^cKD RPE1 cells were analyzed during the first half (day 1–4) (A, B) and the second half (day 5–8) of the screen (C, D). **(A, B, C, D)** The non-CIN over CIN ratios for AUC (A, C) and slope analysis (B, D) were plotted per drug against the P-value. All drugs with log of difference >|0.15|, and P-value < 0.05 are plotted; drugs with P-values < 0.05 after Bonferroni correction are labeled blue.

SKI606 was designed as a tyrosine kinase inhibitor targeting Bcr-Abl (Golas et al, 2003) and Src (Boschelli et al, 2001). However, RPE1 cells do not have the Bcr-Abl fusion, making Src kinase the likely target. To test whether the observed effect of SKI606 on proliferation indeed acts through Src, we next compared the effect of SKI606 with that of another Src inhibitor, SKI-1. We found that SKI-1 displayed a similar synergy with CIN (Figs 4G and S3L; 15–40% more effect than additive; Bliss independence test, P-values 1.5E-3 and 1.1E-3 for first 4 and last 4 d, respectively) as SKI606 (Figs 4H and S3M) in inhibiting proliferation of Mad2^cKD RPE1 cells while having minimal effect on the proliferation of RPE1 control cells. However, as small molecule compounds can have (overlapping) off-target effects, we also wanted to confirm the synergy between Src inhibition and an impaired SAC at the genetic level. For this, we designed three inducible shRNA constructs for Src of which one (shRNA3) yielded a significant knockdown of Src protein levels in RPE1 cells (54% knockdown, Fig S3N). Indeed, we found that RPE1 Src^cKD cells were much more sensitive to the SAC inhibitor reversine than wild-type RPE1 cells (Fig 4I), particularly during the second half of the experiment (days 4–7), similar as observed for Mad2^cKD cells treated with SKI606 (compare Fig 4H with 4I). We, therefore, conclude that pharmaceutical and genetic inhibition of Src is selectively toxic to cells with an impaired SAC.

## The synergy between Mad2 and Src inhibition does not involve impaired DNA damage signaling

Src is an oncogene, a key regulator of cell survival and mitosis (Thomas & Brugge, 1997), an activator of DNA-PK (Dittmann et al, 2008), and a regulator of actin organization (Destaing et al, 2008) and spindle orientation (Nakayama et al, 2012). We, therefore, next asked what the mechanism is between the observed synergy of SAC and Src inhibition in preventing cell proliferation. As CIN leads to DNA damage (Janssen et al, 2011) and Src is involved in activating the DNA damage response via DNA-PK activation (Dittmann et al, 2008), we next asked whether DNA-PK inhibition would reproduce the results observed with Src inhibition. For this, we exposed the cells to a DNA-PK inhibitor at a concentration that significantly increased γ-H2AX foci after gamma radiation, indicating impaired DNA repair (Fig S4A). In this case, we found that DNA-PK inhibition was not synergistically toxic in dox-treated Mad2^cKD cells (Fig S4B). In line with this, another DNA-PK inhibitor that was included in our screen (compound #1463; NU7441) did not show a differential effect between control and CIN RPE1 cells. Finally, we found that 4 Gray of irradiation and SKI606 both decreased proliferation of RPE1 cells as expected, but that SKI606 did not inhibit the growth of irradiated cells more than that of controls, indicating that SKI606-invoked

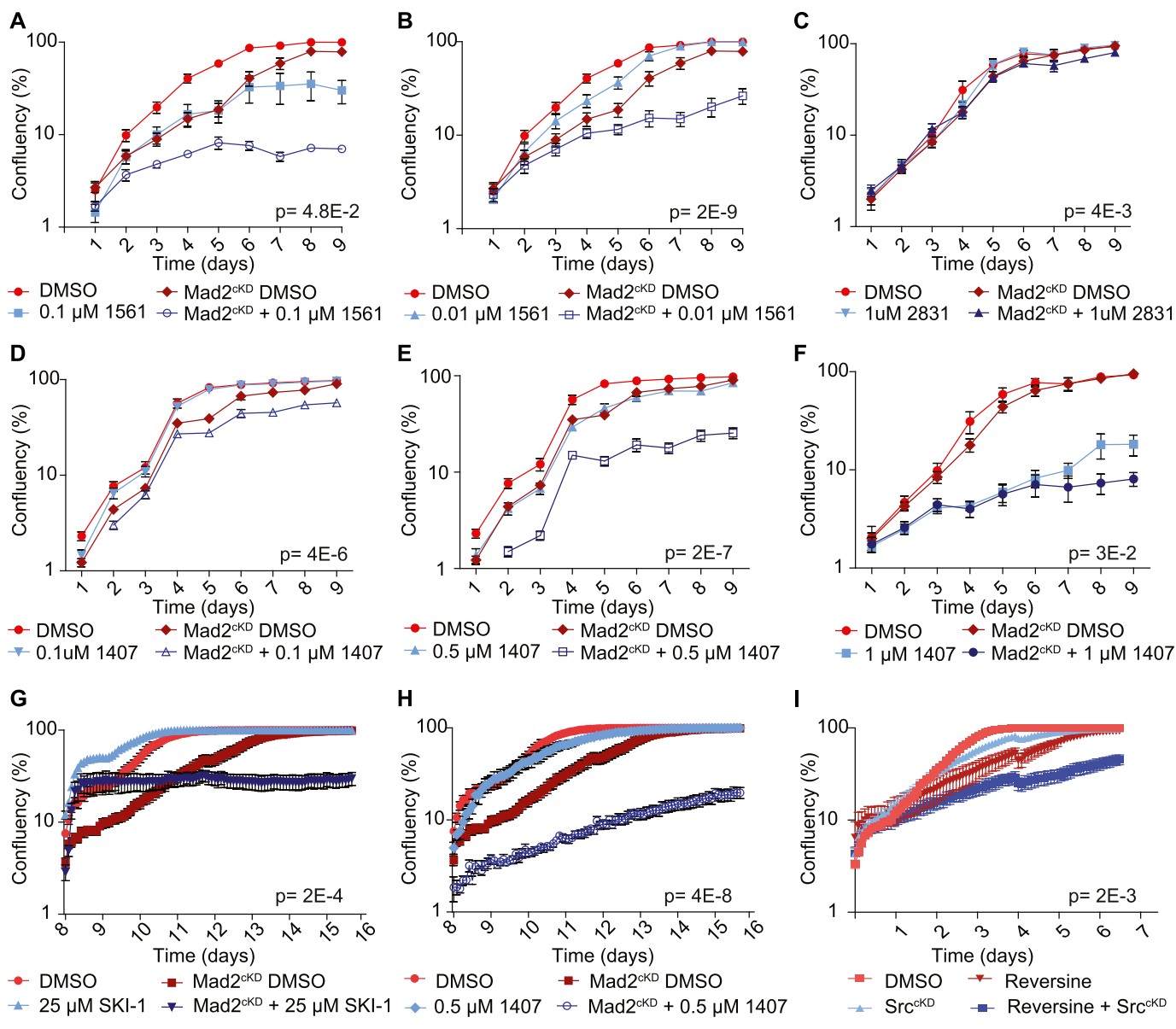

**Figure 4. Validating candidate compounds that selectively target CIN cells.**
**(A, B, C, D, E, F)** Growth curves of control and Mad2$^{cKD}$ RPE1 cells treated with 0.1 $\mu$M (A) and 0.01 $\mu$M (B) compound #1561, (C) 1 $\mu$M compound #2831, (D, E, F) 0.1 $\mu$M, 0.5 $\mu$M, and 1 $\mu$M 1,407, respectively. Data were obtained by sequential daily microscope images and analyzed by FIJI-PHANTAST. Each point is a minimum of three biological replicates, each of which contains three technical replicates. Plotted is log-scaled percentage confluency (cell coverage) over time. Error bars indicate SEM. *P*-values are calculated from paired, one-sided *t* tests of AUC corrected for cell line control. RPE1 DMSO and Mad2$^{cKD}$ DMSO curves shared between (A) and (B), and between (C) and (F) and Fig S3C, and between (D) and (E). **(G, H)** IncuCyte growth curves of control and Mad2$^{cKD}$ RPE1 cells treated with Src inhibitors SKI-1 (G) or compound #1407 (H, SKI606) for day 8–16. **(I)** IncuCyte growth curves for control- and reversine-treated RPE1 cells with and without conditional Src knockdown. All points include data for six technical replicates. Error bars refer to SEM, and *P*-values were calculated from two-sided *t* test of the AUC corrected for cell line controls. Data for DMSO control curves are shared between (G, H) and Fig 5G.

growth inhibition is independent of DNA damage (Fig S4C). We, therefore, conclude that the observed synergy between Mad2 and Src inhibition is not caused by exacerbating DNA damage.

### SKI606 increases CIN in SAC-deficient cells by deregulating MT polymerization rates

Because SKI606 does not appear to target aneuploidy-imposed stresses, nor DNA damage, we next investigated whether SKI606 affects chromosome missegregation rates. For this, we performed

time-lapse imaging experiments with control and Mad2$^{cKD}$ RPE1 cells expressing H2B-GFP and quantified mitotic abnormalities in the presence or absence of SKI606. Interestingly, we found that although SKI606 did not increase CIN in control cells, it did significantly increase CIN in Mad2$^{cKD}$ cells (Fig 5A), increasing the missegregation rates from 46 to 79%.

As a parallel approach, we alleviated the SAC using reversine in RPE1 cells as done for the Src$^{cKD}$ cells (Fig 4I) and found that SKI606 indeed specifically increases chromosome missegregation rates in reversine-treated cells (Fig 5B). We also found that this phenotype

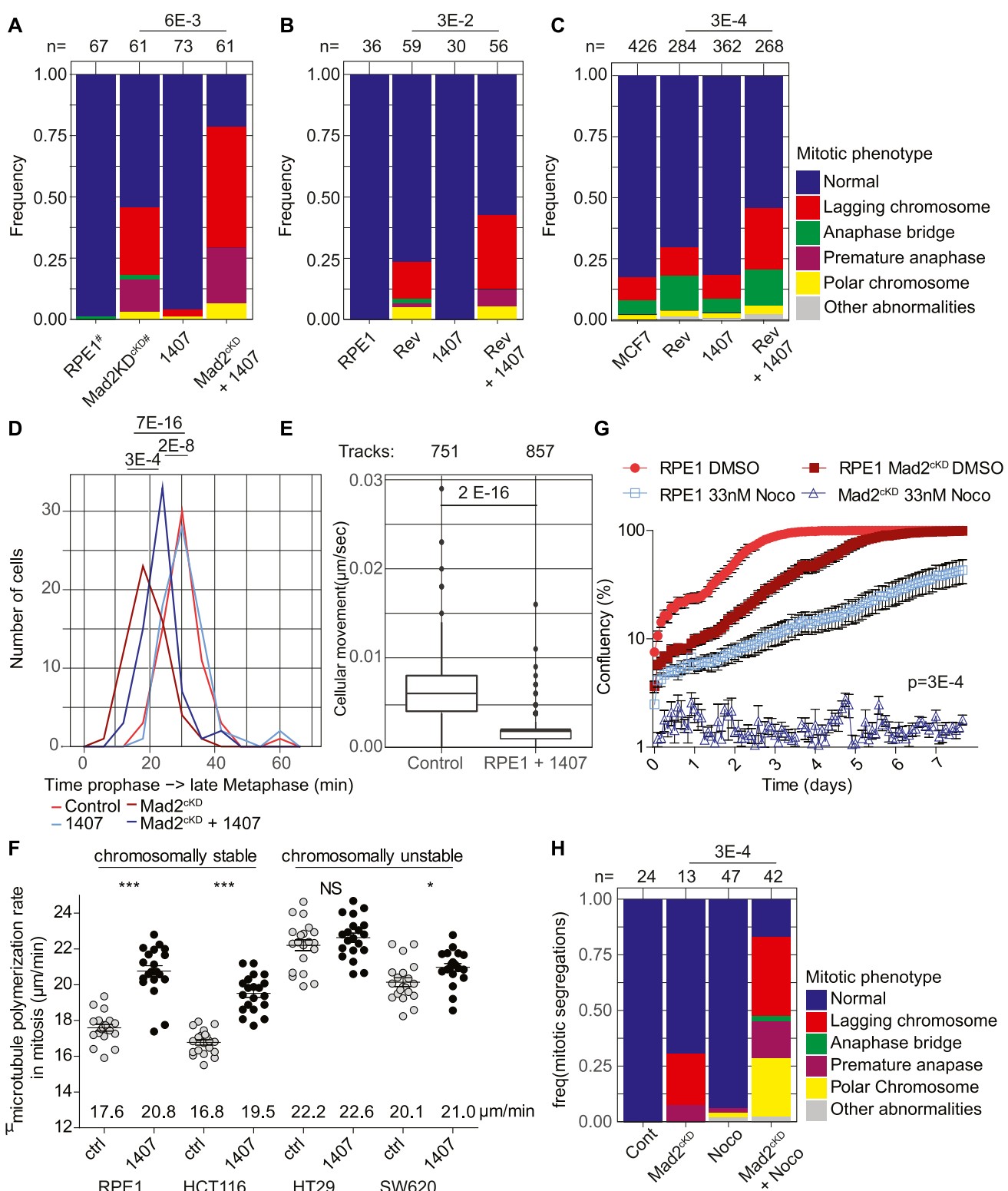

**Figure 5.  1,407 significantly increases CIN in spindle assembly checkpoint (SAC)–deficient cells by altering microtubule (MT) dynamics.**
**(A, B, C)** Frequency of mitotic abnormalities in control and Mad2$^{cKD}$ RPE1 cells with and without 0.5 $\mu$M compound #1407 (A), RPE1 cells with 150 nM reversine with and without 0.5 $\mu$M compound #1407 (B), and MCF7 cells treated with 15 nM reversine and/or 0.5 $\mu$M compound #1407 (C). Data were obtained by time-lapse microscopy imaging and include at least three biological replicates. *P*-values are calculated from chi-squared test. **(D)** Quantification of time from start prophase to late metaphase for control and Mad2$^{cKD}$ RPE1 cells with and without 0.5 $\mu$M compound #1407. At least 29 mitoses were analyzed per condition from a minimum of three time-lapse microscopy experiments. **(E)** Boxplot showing mean cell migration speed ($\mu$m/second) of RPE1 cells with or without 0.5 $\mu$M 1,407. Data include a minimum of three independent

persisted in other cell lines. For instance, SKI606 increased CIN rates of reversine-treated MCF7 breast cancer cells from 30 to 46%, whereas SKI606 did not change CIN rates of MCF7 cells (17–19%) in the absence of reversine (Fig 5C). Together, these observations suggest that Src inhibition exacerbates a CIN phenotype specifically in cells with an impaired SAC.

To further investigate the mechanism underlying the effects of SKI606 on chromosome segregation, we determined whether Src inhibition had an effect on mitotic timing. For this, we compared the mitotic length between control and Mad2$^{cKD}$ RPE1 cells, with and without Src inhibition. Although Mad2 alleviation decreased the time from prophase to metaphase as observed previously (Meraldi et al, 2004), mitotic length again increased when Mad2$^{cKD}$ RPE1 cells were exposed to SKI606 (Fig 5D). This suggests that the increased chromosome missegregation rates in SKI606-treated Mad2$^{cKD}$ cells were not caused by further SAC inhibition and might, therefore, be the result of altered MT dynamics. Mitotic timing of control RPE1 cells was unaffected by SKI606 treatment in line with the absence of a CIN phenotype in SKI606-treated control RPE1 cells. Furthermore, when analyzing the time-lapse data, we also noted that SKI606-treated cells (RPE1 [Fig 5E] and MCF7 cells [Fig S5A]) displayed reduced cell motility, also suggesting an effect of SKI606 on MT dynamics.

Given our results and a known role for Src in spindle orientation (Nakayama et al, 2012) and MT nucleation (Colello et al, 2010), we next investigated the effect of SKI606 on MT dynamics in a number of CIN and non-CIN (cancer) cell lines. For this, we quantified MT dynamics by time-lapse imaging in control- and SKI606-treated cells expressing EB3-GFP, which labels the plus-end tips of MTs and can, therefore, be used to quantify MT dynamics (Stepanova et al, 2003). Taking this approach, we found that SKI606 significantly increased MT polymerization rates in RPE1 as well as in diploid, non-CIN HCT116 cancer cells. Interestingly, we found that SKI606 increased the MT polymerization rates in these non-CIN cell lines to rates comparable with those observed in the CIN cancer cell lines SW620 and HT29 (Fig 5F and Videos 1–8). However, SKI606 treatment failed to further increase MT polymerization rates in HT29 cells, and only had a minor effect on MT polymerization rates in SW620 cells, suggesting that MT polymerization rates had reached their physiological maximum in these lines (Fig 5F and Videos 1–8). Similar as observed for RPE1 and MCF7 cells, we found that SKI606 treatment did not increase chromosome missegregation rates in DMSO-treated HT29 cells (Fig S5B). However, whereas reversine treatment modestly increased CIN rates in HT29 cells as expected, combined SKI606 and reversine treatment failed to increase CIN rates in HT29 cells further (Fig S5B), providing additional proof that SKI606 acts through deregulating MT polymerization rates. Given these results, and as increased MT polymerization rates have previously been shown to drive CIN phenotypes (Ertych et al, 2014), we conclude that SKI606 contributes to a CIN phenotype by altering MT polymerization rates.

## Altering MT dynamics is synergistically toxic with SAC inhibition

To determine whether the synergy between altering MT dynamics and SAC inhibition was specific to SKI606 or would also apply to other MT poisons, we next tested the effect of SAC alleviation with low doses of nocodazole that do not trigger the SAC but only cause increased MT polymerization rates (Ertych et al, 2014). For this, we first determined a nontoxic concentration for long-term (up to 8 d) treatment of nocodazole. Whereas 250, 100, 50, and 25 ng/ml of nocodazole completely inhibited proliferation under these conditions, 10 ng/ml (33 nM) nocodazole was compatible with cell division. Indeed, although 33 nM nocodazole still reduced proliferation of RPE1 control cells, it was significantly more toxic to Mad2$^{cKD}$ RPE1 cells, confirming the synthetic lethality between SAC inhibition and deregulating MT polymerization rates (Figs 5G and S5C, 13% more than additive effect, $P$ = 7.0E-3, Bliss independence test). Also in this setting, the observed synergy between low doses of nocodazole and SAC inhibition coincided with increased chromosome missegregation rates: whereas 33 nM nocodazole provoked mitotic abnormalities in only 6% of control RPE1 cells, 83% of nocodazole-exposed Mad2$^{cKD}$ RPE1 suffered from defective mitoses, compared with 31% in the absence of nocodazole (Fig 5H). Finally, when we combined SKI606 with SAC alleviation in the CIN cell line HT29, in which MT polymerization rates cannot further be increased (Fig 5F [Ertych et al, 2014]), we found that SKI606-imposed Src inhibition was no longer acting synergistically with SAC alleviation in reducing cell numbers (Fig S5D), further indicating that altered MT dynamics is underlying the synergy observed between SKI606 and SAC inhibition. We conclude that altering MT polymerization rates synergizes with SAC inhibition in blocking cell proliferation, thus providing new therapeutic opportunities for cancers in which either the SAC or MT dynamics are disturbed.

# Discussion

CIN and the resulting aneuploidy are hallmark features of cancer cells. As both features discriminate cancer cells from healthy cells, they are promising therapeutic targets. In this study, we explored whether cells exhibiting CIN or stable aneuploidy displayed selective vulnerabilities to particular drugs. As CIN and aneuploidy trigger a number of responses in cells, including, but not limited to proteotoxic stress (Oromendia et al, 2012; Stingele et al, 2012), a deregulated cellular metabolism (Williams et al, 2008; Tang et al, 2011), a DNA damage response (Zhang et al, 2015; MacKenzie et al, 2017), senescence (Andriani et al, 2016), and apoptosis (Ohashi et al, 2015), we selected a small library of Food and Drug Administration-approved drugs or compounds in clinical trials targeting aneuploidy/CIN-related responses.

imaging experiments. $P$-values are calculated using a Wilcox test. **(F)** MT plus end growth rate in mitosis with and without 0.5 $\mu$M compound #1407. Each dot represents the average of 20 MT movements within a cell, 20 cells per condition. **(G)** IncuCyte-based growth curves of control and Mad2$^{cKD}$ RPE1 in the presence or absence of 33 nM nocodazole at days 8–16. AUC is plotted relative to cell line controls, $P$-values are calculated using a Wilcoxon–Mann–Whitney test. Data for DMSO control curves are also used in Fig 4G and H. **(H)** Frequency of mitotic abnormalities in RPE1 cells with or without 0.5 $\mu$M compound #1407 and/or 33 nM nocodazole. Data were obtained by time-lapse microscopy imaging and include at least three biological replicates. $P$-values are calculated by chi-squared test. "n" referrers to the number of mitotic events per condition. "#" refers to that the same data are also used in Fig 2E.

## Aneuploid cells are sensitive to compounds that hyperactivate the cellular metabolism

When we screened for compounds that selectively prevent expansion of aneuploid cells, we found that ZLN005, a transcriptional stimulator of PGC-1α, was significantly more toxic to double-trisomic RPE1 Ts12 Ts5 cells (Stingele et al, 2012) than control cells. PGC-1α is a master regulator of mitochondrial biogenesis and energy metabolism and its activation is, thus, expected to increase mitochondrial respiration. None of the other tested compounds showed reproducible toxicity specific to aneuploid cells. Although somewhat disappointing, it is important to note that we only tested a limited number of compounds (95 in total) in this screen and that large-scale future screens can still reveal new therapeutic vulnerabilities of aneuploid cells. Still, our findings in aneuploid cells correspond well with an earlier study by Tang et al (2011), who identified the energy stress-inducing drug AICAR as a compound that selectively targets aneuploid cells. AICAR activates AMP-activated protein kinase (AMPK) leading to hyperactivation of mitochondrial respiration and, thus, exacerbating metabolic stress (Tang et al, 2011). Interestingly, AMPK acts as an activator of PGC-1α (Jeon, 2016; Tan et al, 2016), and therefore, activation of AMPK through AICAR is expected to phenocopy PGC-1α activation through ZLN005, which is what we find. Therefore, our findings form an important independent confirmation of these earlier findings, and although our findings need to be confirmed in aneuploid cell lines with other karyotypes to rule out karyotype specific effects, they do warrant further research on the molecular mechanism underlying this sensitivity.

Notably, ZLN005 did not emerge as a compound selectively targeting cells with an ongoing CIN phenotype (Mad2$^{cKD}$ RPE1 cells; Fig S3K) although the CIN phenotype in these cells was shown to lead to substantial aneuploidy (Fig 2G). Possibly, aneuploid cells need to adapt to the aneuploid state before becoming sensitive to drugs that exacerbate the cellular metabolisms (Mad2$^{cKD}$ RPE1 cells were only exposed for up to 12 d to a CIN phenotype). Alternatively, as ongoing CIN and aneuploidy trigger (partially) different responses in cells (Bakker et al, 2018 Preprint), they might also display differential vulnerabilities.

## Synthetic lethal interaction between inhibition of the SAC and Src activity

In addition to screening for compounds that selectively prevent accumulation of aneuploid cells, we also screened for compounds that selectively target cells with a CIN phenotype. For the latter, we engineered RPE1 cells in which we could alleviate the SAC in an inducible fashion (Mad2$^{cKD}$ RPE1 cells). This screen identified SKI606, a Src inhibitor as a compound that selectively impairs accumulation of cells with an alleviated SAC. Importantly, we validated the phenotype with another Src inhibitor and confirmed that the effect of the inhibitors is caused by Src inhibition because genetic perturbation of Src by shRNA with SAC inhibition yields the same phenotype as inhibitor treatment with SAC inhibition. Of note, we only succeeded in reducing Src protein levels by approximately twofold using shRNA and failed to engineer Src knockout cell lines using CRISPR engineering (data not shown), which suggests that cells critically rely on

some remaining Src kinase activity for their survival. Therefore, to phenocopy the selective targeting of CIN cells in vivo in future studies, it will be important to titrate drug concentrations well.

We find that Src inhibition increases the chromosome missegregation rate specifically in cells with an impaired mitotic checkpoint. Although Src, to our knowledge, has not been directly implicated with maintaining mitotic fidelity, it has been shown to promote MT nucleation and regrowth (Colello et al, 2010) by binding to γ–tubulin complexes (Kukharskyy et al, 2004). In addition, Src was found to facilitate spindle orientation (Nakayama et al, 2012), and oncogenic v-Src has been associated with cytokinesis failure (Nakayama et al, 2017). This study revealed that Src inhibition results in increased MT polymerization rates, thus exacerbating the CIN phenotype imposed by Mad2 loss. Interestingly, although several Src inhibitors were found to affect tubulin polymerization, this was typically labeled as "dual mechanism of action" rather than a downstream effect of Src inhibition (Liu et al, 2013; Smolinski et al, 2018). Our study suggests that increased MT polymerization rates are a direct consequence of Src inhibition and that, therefore, these effects should be taken into consideration when treating patients with Src inhibitors.

## Mechanism underlying synthetic lethal interaction between SAC alleviation and Src inhibition

What can explain the synergy between SAC alleviation and Src inhibition in killing cells? Our data indicate that Src inhibition leads to increased MT polymerization rates. Importantly, we show that other MT-destabilizing drugs display the same lethal interaction with SAC inhibition, further supporting our hypothesis that the synthetic lethal interaction between SAC and Src inhibition is explained by the role that Src has in regulating MT polymerization rates. But why are SAC-deficient cells specifically vulnerable to deregulated MT dynamics and why does Src inhibition not even impose a modest CIN phenotype upon control RPE1 cells? When MT polymerization rates are increased in cells with a fully functional SAC, this will lead to decreased cell motility and increased kinetochore-MT stability and, thus, hyper-stable kinetochore-MT interactions. In this setting, the SAC will still delay mitosis until all chromosomal abnormalities caused by Src inhibition are resolved. However, when the SAC is also inhibited, it can no longer resolve the hyperstable kinetochore–MT interactions caused by Src inhibition, thus further increasing the frequency of chromosome missegregation events (Fig 6).

Although we have shown in this study that Src inhibition decreases mitotic fidelity specifically in cells with an inhibited spindle checkpoint, we have not further investigated the downstream consequences of the resulting increased CIN rates. It is likely that the growth defects we observed after combined Src and SAC inhibition are caused by a combination of cell cycle arrest and cell death (for a recent review on this topic, see Chunduri & Storchová (2019)) and that whether cells arrest or undergo apoptosis depends on the karyotypes that the aneuploid cells acquired after the CIN insult. As the consequences of CIN can be highly tissue specific (Foijer et al, 2013, 2017), it will be important to further investigate the downstream consequences of altering MT dynamics in combination

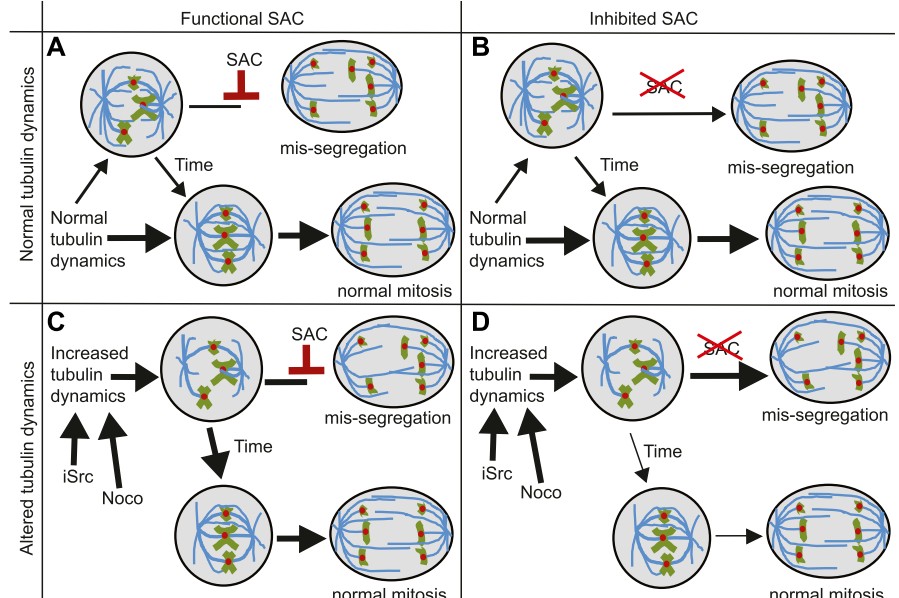

**Figure 6. Proposed mechanism of how increased MT dynamics and SAC inhibition lead to synergistic toxicity by exacerbating the CIN phenotype. (A)** Cells with normal tubulin dynamics and a functional SAC have very low chromosome mis-segregation rates. **(B)** Cells with normal tubulin dynamics but with an alleviated SAC display intermediate chromosome mis-segregation. **(C)** Cells with high tubulin dynamics, but a functioning SAC correct unattached kinetochores before entering anaphase. **(D)** Cells with increased MT dynamics and an alleviated SAC suffer from increased numbers of unaligned chromosomes that are not signaled by the SAC leading to increased rates of chromosome mis-segregation.

with spindle checkpoint inhibition in cell and organoid cultures and in animal models before translating these findings to the clinic.

### Implications for cancer therapy

CIN and aneuploidy are hallmarks of cancer cells, affecting ~70% of all solid cancers (Duijf et al, 2013). Therefore, therapies that exploit this feature might have a broad applicability. Our study suggests that exacerbating CIN in cells with a preexisting CIN phenotype is a powerful strategy to selectively target CIN cells. Indeed, that increasing CIN is a powerful method to target genome instable cancers has been reported by others as well (Kops et al, 2004; Thompson et al, 2010; Silk et al, 2013; Zasadil et al, 2016). One possible explanation for this is that cancer cells tolerate low levels of CIN, until CIN rates exceed a threshold, after which it becomes too toxic for cell survival. Our results indicate that altering MT dynamics to a level that does not affect mitotic fidelity can already act synergistically with SAC defects or inhibitors. This is particularly relevant for cancers in which either the SAC or MT polymerization rates are affected to some extent, as a defect in one process would render the cancer cells extremely sensitive to the interference with the other process, making the therapy much more specific to the cancers cells and thus reducing side effects and long-term toxicity of the treatment.

Alternatively, the synthetic lethal interaction can be exploited to target dividing cancer cells in combination therapy using both drugs at much lower concentrations than when using the drugs as individual agents. Indeed, others have also reported that SAC inhibitors act synergistically with taxanes in killing cells in tissue culture (Janssen et al, 2009; Jemaà et al, 2013; J Tannous et al, 2013; Bargiela-Iparraguirre et al, 2014; Maia et al, 2018) and in vivo in mouse studies (Jemaà et al, 2013; Tannous et al, 2013; Maia et al, 2015, 2018; Wengner et al, 2016). The observed synergy was shown to result from increased CIN (Thompson & Compton, 2008; Janssen et al, 2009). In fact, three clinical trials (NCT03328494, NCT02366949, and NCT03411161) combining Mps1

inhibitors with paclitaxel to target human cancers are currently ongoing (Bayer, 2015; Boston-Pharmaceuticals, 2017; Servier, 2018). In this study, we show that this synergy is not limited to Mps1 and taxanes, and that alternative approaches to alter MT dynamics (such as MT-destabilizing drugs such as vincristine or Src inhibitors) in combination with SAC inhibition can be used to synergistically target CIN cells by significantly increasing CIN. However, before our findings can be taken into the clinic, further validation experiments are required, which should reveal whether Src inhibitors are as effective as other MT polymerization deregulating drugs and whether such drugs indeed act synergistically with SAC inhibitors in targeting cancer cells in vivo.

## Materials and Methods

### Cell culture and compounds

RPE1 and MCF7 (American Type Culture Collection) cells were grown in DMEM supplemented with 10% FBS and 100 units/ml penicillin and 100 μg/ml streptomycin. HT29 cells were growth in McCoy media supplemented with 10% FBS and Pen/Strep as above. Aneuploid cells RPE1 with trisomy 12 and trisomy 5 (Ts12 Ts5) and RPE1 hTert cells were kindly provided by the Storchova laboratory (Stingele et al, 2012). Most drugs used in this study were synthesized by Syncom B.V., except for AZD8055 (Sigma-Aldrich), EPZ015666 (Sigma-Aldrich) and SKI-1 (Abcam). All drugs were dissolved in DMSO (Sigma-Aldrich) and diluted in tissue culture medium as indicated. Used drug concentrations were titrated before the actual screen, starting from an initial drug concentration of 10 μM. If the initial drug concentration of 10 μM was (near-)toxic to wild-type RPE1 cells, the cells were next exposed to 1 μM, 0.1 μM, 10 nM, or 1 nM of the compound, until a concentration was found that was no longer toxic (see Supplemental Data 3 for all initial drug titration growth curves).

## Generation of Mad2<sup>cKD</sup> and Src<sup>cKD</sup> RPE1 cells

Mad2<sup>cKD</sup> RPE1 cells were generated by transducing RPE1 cells with a lentiviral construct targeting human Mad2l1 (5′-GGAAAGAATCAAGGAGG-3′) in a pTRIPZ backbone (Open Biosystems, Cat. no. RHS4696-200677332). The cells were selected in 2 µg/ml puromycin for 48 h and single-cell clones picked. Knockdown efficiency was determined for several clones by Western blot and the clone with the largest Mad2 reduction was used for further experiments. Src<sup>cKD</sup> RPE cells were generated by transducing lentiviral shRNA constructs into wild-type RPE1 cells. For this purpose, Src shRNA constructs were engineered into a doxycycline-inducible pLKO backbone that also contains the Tet operator sequence to allow for conditional activation of the shRNA (Wiederschain et al, 2009). In this vector, we replaced the puromycin resistance gene for a blasticidin resistance gene. In the resulting Tet-pLKO-Blas vector, we cloned the following Src-targeting shRNA sequences using double-stranded oligos ordered from Integrated DNA Technologies (IDT):

shRNA1: 5′-CCGGGCTCGGCTCATTGAAGACAATCTCGAGATTGTCTTCA-ATGAGCCGAGCTTTTTG-3′
shRNA2: 5′-CCGGTCAGAGCGGTTACTGCTCAATCTCGAGATTGAGCA-GTAACCGCTCTGATTTTTG-3′
shRNA3: 5′-CCGGGACAGACCTGTCCTTCAAGAACTCGAGTTCTTGAAGGA-CAGGTCTGTCTTTTTG-3′.

## Western blot

Cells were harvested during the logarithmic growth phase and lysed in Egg Lysis Buffer (150 mM NaCl, 50 mM Hepes, pH 7.5, 5 mM EDTA, and 0.1% NP-40). Protein quantification was performed using Quick Start Bradford assay, the Quick Start BSA standard kit (Bio-Rad Inc.), or the MultiScan Go (Thermo Fisher Scientific). Antibodies used were Actin (#4970; Cell Signaling Technologies), Tubulin (#ab7291; Abcam), Src (#2109; Cell Signaling Technologies), and Mad2 (#610678; BD Bioscience). Secondary antibodies were Anti-Mouse IRDye 680RD (ab216778; Abcam) and Anti-Rabbit IRDye 800CW (ab216773; Abcam). Blots were visualized and quantified using the Odyssey imaging system (Li-Cor).

## Single-cell sequencing

For single-cell sequencing, the cells were harvested, nuclei isolated, and stained with Hoechst using nuclear isolation buffer. Single nuclei were then sorted into 96-well plates using a FACSJazz sorter (BD Bioscience). Single-cell DNA libraries were prepared and sequenced (NextSeq 500; Illumina) with ~1% genomic DNA coverage as described previously (van den Bos et al, 2016). Sequence Binary Alignment maps files were analyzed with AneuFinder version 1.10.2 using the eDivisive analysis model at 1 MB as described elsewhere (Bakker et al, 2016). Sequence Binary Alignment maps files and R script used for analysis are available upon request. Single-cell sequencing data have been deposited at the European Nucleotide Archive under accession number PRJEB33217.

## Metaphase spreads

Cells were cultured with 100 ng/ml Colcemid for 3 h, harvested, incubated in 75 mM KCl for 15 min and fixated in 3:1 methanol: acetic acid. Fixated cells were dropped on glass slides and nuclei visualized using DAPI staining. Metaphase figures were inspected on an Olympus BX43 microscope using a 63× lens. A minimum of 50 karyotyping spreads were counted per condition.

## IncuCyte growth curves

For aneuploid drug screens, 200 RPE1 cells were sorted into each well of 96-well plates by flow cytometry (FACSJazz; BD Bioscience). For the CIN screen and follow-up screens, 1,600 cells were seeded per well in a 96-well plate. For the latter, RPE1 Mad2<sup>cKD</sup> cells were treated with 1 µg/ml doxycycline for a minimum of 3 d before the start of any screen and sorted into wells with 1 µg/ml doxycycline. Each well contained media with drugs at the concentrations listed, and all measurements were performed with technical triplicates for each plate. Cell growth was monitored every 2 h using an IncuCyte Zoom live-cell analysis system (Essen BioScience Ltd.). Drug-containing media were refreshed every 4 d, and for the CIN screen, the cells were passaged 1:8 on day 4. Cell density was quantified using IncuCyte ZOOM 2018A software. Cell confluency of control and CIN cells with drugs were normalized to DMSO-treated cells (RPE1 + DMSO and Mad2<sup>cKD</sup> RPE cells + DMSO, respectively) and calculated using two different approaches: AUC and slope analysis. As for the screens, multiple drugs were tested per plate, the same DMSO control was used (at least one per plate) to compare the effects of the drug-treated cells. The figure legends indicate in which panels the same DMSO control was used.

### AUC IncuCyte gowth curves

The AUC was estimated by taking the sum of the confluences per time-point. These values were then set relative to each individual cell line control AUC. For this, each AUC value was divided by the mean cell line control AUC, that is, RPE1 control, RPE1 double trisomy (Ts12 Ts5), and RPE1 Mad2<sup>cKD</sup> cells were all analyzed as different cell lines so that the relative effect of each perturbation could be compared per cell type. These relative AUC values were then compared between cell lines per drug using a two-sided $t$ test. $P$-values in the screen were corrected for multiple testing using Bonferroni correction (Haynes, 2013).

### Slope analysis

An existing R script to analyze IncuCyte data (Chapman et al, 2016) was modified to find the cutoff point for logarithmic growth. The logarithmic growth cutoff was determined for each drug and cell line combination, and the confluency at the cutoff was taken and divided by the average confluency of the cell line control. Because the slope is defined as the height (confluency) divided by width (time at cutoff), and the cutoff time-point was set to be the same for all DMSO and drug pairs, the cutoff time-points cancel out when setting slope relative to the cell line control. The resulting relative slope values were compared between cell lines per drug type using a $t$ test. The modified IncucyteDRC R package for screen slope analysis is available upon request.

## Bliss independence test for synergistic toxicity between drugs

To validate that the observed effects in the compound screens were synergistic and not only additive, we made use of the Bliss independence test. For this, we calculated the fractional growth inhibition, defined as $1-(AUC^{drug\text{-}treated\ cells}/AUC^{control\ cells})$. Next, we calculated the expected inhibitions, assuming drugs and CIN/aneuploid conditions were additive, using the Bliss independence equation: expected inhibition = Fa + Fb – (Fa*Fb), where Fa is fractional growth inhibition of drug A and Fb was the fractional growth inhibition of either CIN or aneuploid cell conditions. The expected inhibition was compared with the actual growth inhibition; the $P$-value was determined using a $t$ test, and the greater-than-additive toxicity was found by taking the difference between expected and actual fractional inhibition.

## Growth curves analyzed by FIJI PHANTAST

For validation experiments, growth curves were determined from daily microscope images using an Olympus IX51 microscope. For these experiments, 5,300 cells were seeded per well of a 24-well plate. Each well was imaged once a day for a minimum of 7 d. The FIJI package PHANTAST (Jaccard et al, 2014) was used to estimate confluency per well per time-point. PHANTAST settings were $\varepsilon$ = 5 and $\sigma$ ranging between 0.01 and 0.03 depending on cell coverage and confluency accuracy. All measurements include at least three technical replicates and three biological replicates (i.e., a minimum of nine measurements per time point). Growth was plotted using Prism software (GraphPad). As multiple drugs were tested per plate, the same DMSO control was used to compare the effects of the drug-treated cells. The figure legends indicate in which panels the same DMSO control was used. Growth kinetics were normalized identical to IncuCyte measurements.

### AUC PHANTAST growth curves
The AUC was calculated by taking the sum of the confluency at all time-points per condition. This was set relative to cell line growth by dividing each AUC value by the average AUC for the DMSO cell line control for each plate. These values were compared between cell lines per drug with a two-sided $t$ test.

### Slope analysis FIJI growth curves
Growth curves were plotted on a log scale and the logarithmic growth cutoff point was estimated manually for each condition. To calculate the slope, the negative log of the confluency at that time-point was divided by the cutoff day: $-\log(confluency_{cutoff})/T_{cutoff}$. This was divided by the average slope of the cell line control to compensate for cell line growth differences. The replicates of the relative slope values were compared between cell lines per condition.

## Live cell imaging and CIN analysis

RPE1, Mad2$^{cKD}$ RPE1, MCF7, and HT29 cells expressing H2B-GFP were treated as indicated and imaged on a DeltaVision microscope (Applied Precision Ltd.). Interphase phenotypes were analyzed by quantifying nuclear morphology. Mitotic abnormalities were manually quantified from overnight live cell imaging movies. Measurements include at least three biological replicates, and numbers of cells quantified are indicated in the text. A chi-squared test was used to test whether differences between conditions were significant. Mitotic time was analyzed by calculating the time-points between the first sign of DNA condensation to the last point before anaphase, and from the first anaphase time-point to the time-point at complete DNA de-condensation from time-lapse imaging data.

## Cell motility assay

To quantify cell motility, time-lapse movies were analyzed using FIJI TrackMate (Tinevez et al, 2017). A minimum of 15 overnight imaging movies were used per condition, including at least three biological replicates. TrackMate input conditions were optimized for each cell type. RPE1 nuclear diameter was set at 20 $\mu$M, whereas MCF7 nuclear diameter was set at 15 $\mu$M. Track statistics per condition were combined and Track speed was plotted in R ggPlot2 (Wickham, 2016). Differences were calculated with two-sided $t$ tests.

## MT movement analysis

MT plus end assembly rates were determined by tracking EB3-GFP protein (vector kindly provided by Linda Wordeman) in live cell microscopy experiments as in Ertych et al (2014). Average assembly rates (micrometer per minute) were calculated based on data retrieved for 20 individual MTs per cell that were randomly selected. A total of 20 cells were analyzed from three independent experiments. Significance was assessed using a two-sided, unpaired $t$ test.

# Supplementary Information

# Acknowledgements

We are grateful to the members of the Foijer and Bruggeman labs and to the members of the PloidyNet consortium. This work was supported by the European Union FP7 Marie Curie Innovative Training Network grant PloidyNet (607722) to F Foijer and a Dutch Cancer Society project grant (2015-RUG-7833) to F Foijer.

## Author Contributions

KM Schukken: conceptualization, data curation, software, formal analysis, investigation, methodology, and writing—original draft.
Y-C Lin: investigation, methodology, and writing—review and editing.
PL Bakker: investigation and methodology.
M Schubert: software, supervision, and writing—review and editing.
SF Preuss: resources, investigation, and methodology.
JE Simon: resources and methodology.
H van den Bos: resources, data curation, and investigation.
Z Storchova: resources and writing—review and editing.

M Colomé-Tatché: funding acquisition, writing–review and editing, and supervision.

H Bastians: data curation, supervision, and writing—review and editing.

DCJ Spierings: formal analysis, supervision, and methodology.

F Foijer: conceptualization, data curation, formal analysis, supervision, funding acquisition, project administration, and writing—original draft, review, and editing.

## Conflict of Interest Statement

The authors declare that they have no conflict of interest.

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
