## [Reviewer comments · Life Science Alliance]

Altering microtubule dynamics is synergistically toxic with spindle assembly checkpoint inhibition

Klaske Schukken, Yu-Chih Lin, Petra Bakker, Michael Schubert, Stephanie Preuss, Judith Simon, Hilda van den Bos, Zuzana Storchova, Holger Bastians, Diana Spierings, and Floris Foijer

DOI: 10.26508/lsa.201900499

Corresponding author(s): Floris Foijer, University Medical Center Groningen

Review timeline:

Submission Date:	2019-07-24
Editorial Decision:	2019-08-11
Revision Received:	2020-01-02
Editorial Decision:	2020-01-03
Revision Received:	2020-01-07
Accepted:	2020-01-08

Scientific Editor: Andrea Leibfried

Transaction Report:

August 11, 2019

Re: Life Science Alliance manuscript #LSA-2019-00499-T

Dr. Floris Fojjer
University Medical Center Groningen
European Institute for the Biology of Aging
Antonius Deusinglaan 1
Groningen 9713AV
Netherlands

Dear Dr. Fojjer,

Thank you for submitting your manuscript entitled "Altering microtubule dynamics is synergistically toxic with inhibition of the spindle checkpoint" to Life Science Alliance. The manuscript was assessed by expert reviewers, whose comments are appended to this letter.

As you will see, the reviewers think that your findings are important and they provide constructive input on how to further strengthen your study. We would thus like to invite you to provide a revised version of your manuscript, addressing the individual reviewer points raised / suggestions made. This seems rather straightforward, but please do get in touch in case you would like to discuss an individual revision point further.

Thank you for this interesting contribution to Life Science Alliance. We are looking forward to receiving your revised manuscript.

Sincerely,

Andrea Leibfried, PhD
Executive Editor
Life Science Alliance
Meyerohofstr. 1
69117 Heidelberg, Germany
t +49 6221 8891 502

e.a.leibfried@life-science-alliance.org
www.life-science-alliance.org

B. MANUSCRIPT ORGANIZATION AND FORMATTING:

Reviewer #1 (Comments to the Authors (Required)):

In this manuscript, Schukken et al perform targeted screens to identify drugs that are FDA approved or in clinical trials that are preferentially toxic to RPE cells that are aneuploid due to one extra copy of chromosomes 5 and 12 or are CIN due to Mad2 conditional knockdown (Mad2cKD). Of 95 molecules tested, only ZLN005, a transcriptional regulator of PGC-1 α that regulates energy metabolism, was significantly more toxic to the aneuploid RPE cells. Of 58 compounds tested, the Src inhibitor SKI606 showed the largest growth inhibitory effect on Mad2cKD cells over 8 days of growth. While SKI606 did not increase CIN in diploid RPE cells, it substantially increased lagging chromosomes in Mad2cKD cells and in cells treated with the Mps1 inhibitor reversine. SKI606 significantly increased microtubule polymerization rates in chromosomally stable RPE and HCT116 cells as well as CIN SW620 cells, but not CIN HT29 cells, which had the highest baseline rate of microtubule polymerization. Consistent with an effect on microtubule dynamics being the feature of SKI606 that contributed to CIN, it did not increase CIN when combined with reversine in HT29 cells

where it did not impact microtubule polymerization rates. Mad2cKD also showed synthetic lethality with a low dose of the microtubule poison nocodazole. The authors conclude that inhibiting Src increases microtubule polymerization rates and that deregulating microtubule polymerization rates is particularly toxic to cells with a defective SAC.

Since aneuploidy and CIN are hallmarks of tumors, identifying treatments that selectively target these characteristics is of interest. The finding that SKI606 is synergistically lethal with SAC impairment due to effects on microtubule dynamics that further increase CIN is well supported. Although the work would be strengthened by genetic depletion of Src, the inclusion of an experiment with a second Src inhibitor (SKI-1) strengthens the conclusion that the synergy is an on-target effect of SKI606. The conclusion that SAC defects are synergistically lethal with defects in microtubule dynamics is consistent with the fact that the SAC genes (including Mad2) were originally identified because their mutation caused hypersensitivity to microtubule poisons in yeast.

Minor concerns

For most of the growth curves, the control cells appear to become 100% confluent at about day 6. Although the Mad2cKD cells often show a growth defect at earlier time points, they are able to catch up to the control cells the last few days of the experiment because the control cells have already reached maximal confluency. Since the cell confluency in the presence of drug was normalized to confluency in DMSO, this would be expected to make the impact on the Mad2cKD cells appear larger, which should be discussed.

The indication at the top of page 11 that increased microtubule polymerization rates lead to decreased kinetochore-microtubule stability is surprising. I would anticipate increased microtubule polymerization rates would lead to increased kinetochore-microtubule stability (ie hyper stable kinetochore microtubules). This would be consistent with the increased rate of lagging chromosomes observed after SKI606 treated Mad2cKD or reversine treated cells.

It would be helpful to add supplemental movies showing the microtubule polymerization rates with and without SKI606.

A reference to Bakhom et al, NCB 2009 should be added to the statement that "altered microtubule dynamics are another source of CIN" on page 3.

I don't believe the review cited by Gordon et al, 2012 indicates that complete loss of SAC function is rare in human cancer, I believe it says that SAC impairment is rare. I'm not sure it's been shown that complete loss of the SAC occurs.

When citing the clinical trials using Mps1 inhibitors with paclitaxel, it would be helpful to cite the clinical trial identifiers (NCT03328494, NCT02366949, NCT03411161).

Reviewer #2 (Comments to the Authors (Required)):

The paper by Schukken et al describes two anti-cancer drug screens, one in an aneuploid cell line and one in a chromosomally unstable cell line. In the first screen, they identify a drug that affects energy metabolism as synthetically-lethal with the gain of chromosomes 5 and 12 in RPE1 cells. As this result largely recapitulates a prior report from the Amon lab, they do not characterize it further. In the second screen, they identify the drug SKI-606 as synthetically-lethal with MAD2-knockdown in RPE1 cells. SKI-606 increases the rate of microtubule polymerization, and they show that

other treatments that also affect microtubules are synthetically lethal with RPE1 CIN.

Overall, this study is well-conducted and the data is appropriately analyzed. It will add to the literature on genomic instability in cancer, and potential ways to target unstable cells. I recommend publication with a few minor text edits:

Overall: the authors should address the observation that SKI-606 is a very promiscuous drug (<http://lincs.hms.harvard.edu/db/datasets/20184/results>). Showing consistency with a second drug is good, but not as good as a CRISPR experiment. However, that experiment is not necessary for this paper. Instead, the authors can simply discuss the possibility that the lethality phenotypes are caused by some other effect of the compound, as they have minimal evidence demonstrating that SRC inhibition is its cause. (Could the drug itself target microtubules?)

Page 3: there is a random "(28)" in the middle of the text - maybe an improperly-formatted reference.

Page 3: the authors cite the same Tang 2011 paper twice.

Page 5: RPE1 is near-diploid, not diploid, as it has a large aneuploidy on chromosome 5.

Page 6: Mad2cKO should be MAD2cKD.

Page 6: 45% of knockdown cells displayed "multiple aneuploidies per cell". Are there really multiple aneuploidies in every cell? It is hard to tell in Figure 2. If so, why were no cells recovered with single aneuploidies?

Page 9: "its activation is thus expected to increase the cellular metabolism." What does increase the cellular metabolism mean? Metabolism of what? A more specific phrase could be used.

Page 10: explaining why ZLN005 wasn't synthetically-lethal with CIN, the authors should mention the possibility that it is synthetically lethal with aneuploidy for Chr5 or Chr12, but not other aneuploidies

Reviewer #3 (Comments to the Authors (Required)):

In their manuscript, Schukken et al. performed two chemical screens, one to identify drugs that target aneuploid cells, and another to find drugs that are toxic to cells that show CIN (chromosomal instability). They found one drug (ZLN005) that inhibits the proliferation of an aneuploid cell line and several drugs that preferentially inhibit the proliferation of cells where Mad2 has been knock-downed and show a CIN phenotype. The experiments are well-designed and the findings are novel. Importantly, the identification of new strategies to target aneuploid cells and cells with ongoing CIN is important for cancer research.

A few suggestions:

The authors show that ZLN005 inhibits the proliferation of one aneuploid cell line (Trisomy 5 and 12). To claim that ZLN005 inhibits the proliferation of aneuploid cells in general, other aneuploid cell lines could be tested.

In the second screen, they identified a drug that inhibits Src kinase to selectively inhibit the proliferation of cells where Mad2 levels are low due to its knockdown. The authors suggest that Src1 inhibition exacerbates a CIN phenotype by increasing microtubule

polymerization rates. It is unclear how increasing microtubule polymerization rates promote chromosome missegregation.

The cancer cell lines HT29 and SW620 which show high CIN are not affected by Src inhibition. Therefore, SKI606 does not target CIN or aneuploid cells, it only inhibits the proliferation of cells where Mad2 has been knockdown. Do the other drugs identified in the screen (AZD8055 or EPZ015666) target CIN cells?

To validate the specificity of Src inhibition, cells were treated with another inhibitor of Src kinase (SKI-1). Indeed, they show that this inhibitor also inhibits the proliferation of Mad2-kD cells. However the growth curves profiles of cells treated with these two drugs look very different (4G vs. 4H) pointing toward different mechanisms of action. An independent method should be used to validate that indeed that inhibition of Src kinase activity is responsible for the selective inhibition of proliferation of Mad2-kD cells.

Confusingly, Src inhibition does not affect the proliferation or cause a mitotic phenotype in RPE1 or MCF7 (Figures 4A and 4C), yet this drug affects cell motility and microtubule polymerization. Do these data suggest that altering microtubule dynamics do not affect mitosis?

The graphs showing the effects of the drugs (1A-D, 3A-D) are very confusing. It is not clear what "log2 difference" reports. Does log2 difference equal to 1 mean that the area under the curve of control minus cells treated with the drug is 2? Do the authors mean log2 of the ratios? If this is the case a value of 1 means that the AUC has increased 2 fold upon drug treatment? Or decrease by 50%?

The axis of several figures is mislabeled log (%Confluency). A log10? of 100% would be 2.

1407 toxic only after 5-8 days. Is the reason for this result since it takes several days for the cells to become highly aneuploid? Does 1407 treatment affect the karyotype of cells?

The authors use the word killing throughout the manuscript. However, lower confluency could be due to inhibition of proliferation not due to increased cell death. The levels of cell death or apoptosis were not reported.

In the third paragraph of the introduction, what does CIN (28) mean?

Reviewer #1 (Comments to the Authors (Required)):

In this manuscript, Schukken et al perform targeted screens to identify drugs that are FDA approved or in clinical trials that are preferentially toxic to RPE cells that are aneuploid due to one extra copy of chromosomes 5 and 12 or are CIN due to Mad2 conditional knockdown (Mad2cKD). Of 95 molecules tested, only ZLN005, a transcriptional regulator of PGC-1 α that regulates energy metabolism, was significantly more toxic to the aneuploid RPE cells. Of 58 compounds tested, the Src inhibitor SKI606 showed the largest growth inhibitory effect on Mad2cKD cells over 8 days of growth. While SKI606 did not increase CIN in diploid RPE cells, it substantially increased lagging chromosomes in Mad2cKD cells and in cells treated with the Mps1 inhibitor reversine. SKI606 significantly increased microtubule polymerization rates in chromosomally stable RPE and HCT116 cells as well as CIN SW620 cells, but not CIN HT29 cells, which had the highest baseline rate of microtubule polymerization. Consistent with an effect on microtubule dynamics being the feature of SKI606 that contributed to CIN, it did not increase CIN when combined with reversine in HT29 cells where it did not impact microtubule polymerization rates. Mad2cKD also showed synthetic lethality with a low dose of the microtubule poison nocodazole. The authors conclude that inhibiting Src increases microtubule polymerization rates and that deregulating microtubule polymerization rates is particularly toxic to cells with a defective SAC.

Since aneuploidy and CIN are hallmarks of tumors, identifying treatments that selectively target these characteristics is of interest. The finding that SKI606 is synergistically lethal with SAC impairment due to effects on microtubule dynamics that further increase CIN is well supported. Although the work would be strengthened by genetic depletion of Src, the inclusion of an experiment with a second Src inhibitor (SKI-1) strengthens the conclusion that the synergy is an on-target effect of SKI606. The conclusion that SAC defects are synergistically lethal with defects in microtubule dynamics is consistent with the fact that the SAC genes (including Mad2) were originally identified because their mutation caused hypersensitivity to microtubule poisons in yeast.

Minor concerns

For most of the growth curves, the control cells appear to become 100% confluent at about day 6. Although the Mad2cKD cells often show a growth defect at earlier time points, they are able to catch up to the control cells the last few days of the experiment because the control cells have already reached maximal confluency. Since the cell confluency in the presence of drug was normalized to confluency in DMSO, this would be expected to make the impact on the Mad2cKD cells appear larger, which should be discussed.

>Many thanks for this comment. We apologize for the confusion. All growth curves were normalized to their 'unperturbed' control. We agree this was not completely clear from the material and methods. We adapted the text to read: *"The AUC was estimated by taking the sum of the confluences per time-point. These values were then set relative to each individual cell line control area under the curve. For this, each AUC value was divided by the mean cell line control AUC:*

i.e. RPE1 control, RPE1 double trisomy (Ts12 Ts5) and RPE1 Mad2cKD cells were all analyzed as different cell lines so that the relative effect of each perturbation could be compared per cell type.” We also clarified this throughout the text where applicable. We hope this is now much better explained.

The indication at the top of page 11 that increased microtubule polymerization rates lead to decreased kinetochore-microtubule stability is surprising. I would anticipate increased microtubule polymerization rates would lead to increased kinetochore-microtubule stability (ie hyper stable kinetochore microtubules). This would be consistent with the increased rate of lagging chromosomes observed after SKI606 treated Mad2cKD or reversine treated cells.

> Thanks for this remark. The reviewer is absolutely correct, as this was shown e.g. by our collaborator Holger Bastians (Ertych et al, Nat Cell Biol, 16 (8), 779-91) and Bakhoum et al in Nat Cell Bio in 2009. We corrected this and the next now reads: *“Therefore, if MT polymerization rates are increased, thus leading to increased kinetochore-MT stability and thus hyper-stable kinetochore-MT interactions, the SAC will still delay mitosis until all chromosomes are properly attached.”*

It would be helpful to add supplemental movies showing the microtubule polymerization rates with and without SKI606.

> We will provide these movies to the editor to make them available online if the manuscript were to be accepted for publication, but the postdoc who made the movies and analyzed them is travelling until the second half of January.

A reference to Bakhoum et al, NCB 2009 should be added to the statement that "altered microtubule dynamics are another source of CIN" on page 3.

> We have added this reference. Apologies for our oversight.

I don't believe the review cited by Gordon et al, 2012 indicates that complete loss of SAC function is rare in human cancer, I believe it says that SAC impairment is rare. I'm not sure it's been shown that complete loss of the SAC occurs.

> We agree with this notion and altered the text as follows: *“While SAC impairment is rare in human cancer (Gordon et al, 2012), many cancers show signs of a partly impaired SAC, for instance as a result of increased expression of proteins with a direct role in the SAC or their regulators, such as Rb mutations that lead to increased expression of Mad2 and thus provoke a CIN phenotype (Pfau & Amon, 2012).”*

When citing the clinical trials using Mps1 inhibitors with paclitaxel, it would be helpful to cite the clinical trial identifiers (NCT03328494, NCT02366949, NCT03411161).

> Many thanks for this suggestion. We have incorporated the trial identifiers in the text. It now reads: *“In fact, three clinical trials (NCT03328494, NCT02366949, NCT03411161) combining Mps1 inhibitors with Paclitaxel to target human cancers*

are currently ongoing (Boston-Pharmaceuticals, 2017; Servier, 2018; Bayer, 2015).”

Reviewer #2 (Comments to the Authors (Required)):

The paper by Schukken et al describes two anti-cancer drug screens, one in an aneuploid cell line and one in a chromosomally unstable cell line. In the first screen, they identify a drug that affects energy metabolism as synthetically-lethal with the gain of chromosomes 5 and 12 in RPE1 cells. As this result largely recapitulates a prior report from the Amon lab, they do not characterize it further. In the second screen, they identify the drug SKI-606 as synthetically-lethal with MAD2-knockdown in RPE1 cells. SKI-606 increases the rate of microtubule polymerization, and they show that other treatments that also affect microtubules are synthetically lethal with RPE1 CIN.

Overall, this study is well-conducted and the data is appropriately analyzed. It will add to the literature on genomic instability in cancer, and potential ways to target unstable cells. I recommend publication with a few minor text edits:

Overall: the authors should address the observation that SKI-606 is a very promiscuous drug (<http://lincs.hms.harvard.edu/db/datasets/20184/results>). Showing consistency with a second drug is good, but not as good as a CRISPR experiment. However, that experiment is not necessary for this paper. Instead, the authors can simply discuss the possibility that the lethality phenotypes are caused by some other effect of the compound, as they have minimal evidence demonstrating that SRC inhibition is its cause. (Could the drug itself target microtubules?)

>Many thanks for this suggestion. While this reviewer does not ask for genetic perturbation of Src, we fully agree this is an important point as many drugs (even when testing multiple) have off-target effects. We therefore engineered inducible shRNA constructs targeting Src and now show that genetic perturbation of Src phenocopies the effect of SKI-606. We added the following text to the results: *“However, as small molecule compounds can have (overlapping) off-target effects, we also wanted to confirm the synergy between Src inhibition and an impaired SAC at the genetic level. For this, we designed 3 inducible shRNA constructs for Src of which one (shRNA3) yielded a significant knockdown of Src protein levels in RPE1 cells (54% knockdown, Sup. Fig. 3N). Indeed, we found that RPE1 SrcKD cells were much more sensitive to the SAC inhibitor Reversine than wildtype RPE1 cells (Fig. 4I), particularly during the second half of the experiment (days 4-7), similar as observed for Mad2cKD cells treated with SKI606 (compare Fig. 4H to 4I). We therefore conclude that pharmaceutical as well as genetic inhibition of Src is selectively toxic to cells with an impaired spindle assembly checkpoint.”* We added panel I to Figure 4 and panel N to Sup. Figure 3 showing the data and added some text to the discussion as well: *“Importantly, we validated the phenotype with another Src inhibitor and confirmed that the effect of the inhibitors is caused by Src inhibition, since genetic perturbation of Src by shRNA with SAC inhibition yields the same phenotype as inhibitor treatment with SAC inhibition. Of note, we only succeeded in reducing Src protein levels by approximately 2-fold using shRNA and failed to engineer Src knockout cell lines using CRISPR engineering (data not*

shown), which suggests that cells critically rely on some remaining Src kinase activity for their survival. Therefore, in order to phenocopy the selective targeting of CIN cells in vivo in future studies, it will be important to titrate drug concentrations well." Note that we also tried engineering Src KO lines using CRISPR engineering, a technique that works very efficiently in the lab. In this case, we tested 3 different guideRNAs in 6 independent transfection rounds which yielded no Src KO cell lines, strongly suggesting that Src loss is toxic. However, with this new data, we are convinced that the synergy between SAC inhibition and SKI-606 is indeed explained by Src inhibition.

Page 3: there is a random "(28)" in the middle of the text - maybe an improperly-formatted reference.

>Thanks for noting that, we have corrected this.

Page 3: the authors cite the same Tang 2011 paper twice.

>Thanks for noting this, we have corrected this.

Page 5: RPE1 is near-diploid, not diploid, as it has a large aneuploidy on chromosome 5.

>Indeed, the reviewer is correct to state that RPE1 cells are not diploid, but near-diploid. They carry a large amplification of chr. 10, as can be seen in Fig. 2F. We corrected this in the text, which now reads: "*For this, we exposed wildtype RPE1 cells (a near-diploid non-cancer cell line derived from retinal epithelium (Soto et al, 2017)) to decreasing concentrations of the drugs, starting at 10 μ M for all compounds, and compared cell proliferation of drug-exposed cells to proliferation of DMSO-treated cells over a period of 7 days.*" and "*To quantify differences between near-diploid and aneuploid RPE1 cells, we compared the area under the curve (AUC) as a measure of cumulative cell growth (Fig. 1A, B) and the slope of the logarithmic growth as a measure for the proliferation rate (Fig. 1C, D), also see Materials and Methods.*"

Page 6: Mad2cKO should be MAD2cKD.

>Many thanks for this. We corrected this. The text now reads: "*While control RPE1 cells show little aneuploidy (2 out of 114 cells sequenced) except for a known structural abnormality for chr. 10 (Fig. 2F, and (Worrall et al, 2018)), 45% of dox-treated Mad2cKD cells displayed multiple aneuploidies per cell (76 out of 169 cells, Fig. 2G) within 5 days after induction of the Mad2 shRNA, confirming a substantial CIN phenotype.*"

Page 6: 45% of knockdown cells displayed "multiple aneuploidies per cell". Are there really multiple aneuploidies in every cell? It is hard to tell in Figure 2. If so, why were no cells recovered with single aneuploidies?

>Thanks for bringing this up. We looked carefully at the SCS data again and find that Mad2 knockdown leads to cells with single and multiple aneusomies per cell. We rephrased the text to read: "*While control RPE1 cells show little aneuploidy (2*

out of 114 cells sequenced) except for a known structural abnormality for chr. 10 (Fig. 2F, and (Worrall et al, 2018)), 45% of dox-treated Mad2cKD cells displayed one to few aneusomies per cell (76 out of 169 cells, Fig. 2G) within 5 days after induction of the Mad2 shRNA, confirming a substantial CIN phenotype.”

Page 9: "its activation is thus expected to increase the cellular metabolism." What does increase the cellular metabolism mean? Metabolism of what? A more specific phrase could be used.

>We agree that this statement was too vague. We extended the discussion section to make this statement more specific and link it better to earlier findings. The paragraph now reads: *“When we screened for compounds that selectively prevent expansion of aneuploid cells, we found that ZLN005, a transcriptional stimulator of PGC-1 α , was significantly more toxic to double-trisomic RPE1 Ts12 Ts5 cells (Stingele et al, 2012) than control cells. PGC-1 α is a master regulator of mitochondrial biogenesis and energy metabolism and its activation is thus expected to increase mitochondrial respiration. None of the other tested compounds showed reproducible toxicity specific to aneuploid cells. While somewhat disappointing, it is important to note that we only tested a limited number of compounds (95 in total) in this screen and that large-scale future screens can still reveal new therapeutic vulnerabilities of aneuploid cells. Still, our findings in aneuploid cells correspond well with an earlier study by Tang et al (Tang et al, 2011), who identified the energy stress-inducing drug AICAR as a compound that selectively targets aneuploid cells. AICAR activates AMP-activated protein kinase (AMPK) leading to hyperactivation of mitochondrial respiration and thus exacerbating metabolic stress (Tang et al, 2011). Interestingly, AMPK acts as an activator of PGC-1 α (Jeon, 2016; Tan et al, 2016) and therefore activation of AMPK through AICAR is expected to phenocopy PGC-1 α activation through ZLN005, which is what we find. Therefore, our findings form an important independent confirmation of these earlier findings, and while our findings need to be confirmed in aneuploid cell lines with other karyotypes to rule out karyotype specific effects, they do warrant further research on the molecular mechanism underlying this sensitivity.”*

Page 10: explaining why ZLN005 wasn't synthetically-lethal with CIN, the authors should mention the possibility that it is synthetically lethal with aneuploidy for Chr5 or Chr12, but not other aneuploidies

> We agree that our findings need to be confirmed in other karyotypes, which is now added to the last sentence of this paragraph: *“Therefore, our findings form an important independent confirmation of these earlier findings, and while our findings need to be confirmed in aneuploid cell lines with other karyotypes to rule out karyotype specific effects, they do warrant further research on the molecular mechanism underlying this sensitivity.”*

Reviewer #3 (Comments to the Authors (Required)):

In their manuscript, Schukken et al. performed two chemical screens, one to identify drugs that target aneuploid cells, and another to find drugs that are toxic to cells that show CIN (chromosomal instability). They found one drug (ZLN005) that inhibits the proliferation of an aneuploid cell line and several drugs that

preferentially inhibit the proliferation of cells where Mad2 has been knock-downed and show a CIN phenotype. The experiments are well-designed and the findings are novel. Importantly, the identification of new strategies to target aneuploid cells and cells with ongoing CIN is important for cancer research.

A few suggestions:

The authors show that ZLN005 inhibits the proliferation of one aneuploid cell line (Trisomy 5 and 12). To claim that ZLN005 inhibits the proliferation of aneuploid cells in general, other aneuploid cell lines could be tested.

>Many thanks for this suggestion. We fully agree that in order to make this statement stronger, we should have tested ZLN005 in RPE1 lines with other aneuploidies. As the focus of the manuscript was much more on the compounds that kill CIN cells, we focused on making the validation of our Src findings stronger. However, we have discussed our findings in stable aneuploid cells more extensively in the discussion section and also placed them in the perspective that our findings need to be confirmed for other karyotypes: *“When we screened for compounds that selectively prevent expansion of aneuploid cells, we found that ZLN005, a transcriptional stimulator of PGC-1 α , was significantly more toxic to double-trisomic RPE1 Ts12 Ts5 cells (Stingele et al, 2012) than control cells. PGC-1 α is a master regulator of mitochondrial biogenesis and energy metabolism and its activation is thus expected to increase mitochondrial respiration. None of the other tested compounds showed reproducible toxicity specific to aneuploid cells. While somewhat disappointing, it is important to note that we only tested a limited number of compounds (95 in total) in this screen and that large-scale future screens can still reveal new therapeutic vulnerabilities of aneuploid cells. Still, our findings in aneuploid cells correspond well with an earlier study by Tang et al (Tang et al, 2011), who identified the energy stress-inducing drug AICAR as a compound that selectively targets aneuploid cells. AICAR activates AMP-activated protein kinase (AMPK) leading to hyperactivation of mitochondrial respiration and thus exacerbating metabolic stress (Tang et al, 2011). Interestingly, AMPK acts as an activator of PGC-1 α (Jeon, 2016; Tan et al, 2016) and therefore activation of AMPK through AICAR is expected to phenocopy PGC-1 α activation through ZLN005, which is what we find. Therefore, our findings form an important independent confirmation of these earlier findings, and while our findings need to be confirmed in aneuploid cell lines with other karyotypes to rule out karyotype specific effects, they do warrant further research on the molecular mechanism underlying this sensitivity.”*

In the second screen, they identified a drug that inhibits Src kinase to selectively inhibit the proliferation of cells where Mad2 levels are low due to its knockdown. The authors suggest that Src1 inhibition exacerbates a CIN phenotype by increasing microtubule polymerization rates. It is unclear how increasing microtubule polymerization rates promote chromosome missegregation.

>This point is related to the second point of reviewer 1. In the first version of the manuscript, we stated that increased MT polymerization rates leads to decreased kinetochore-MT stability. This was incorrect and confusing. We meant to say that this leads to increased kinetochore-MT stability and thus hyperstable kinetochore-

MT interactions. Such hyperstable interactions have previously been shown to lead to CIN, for instance by the labs of Holger Bastians (Bastians (Ertych et al, Nat Cell Biol, 16 (8), 779-91) and Duane Compton (Bakhoun et al, Nat. Cell Biol. 11: 27–35). We rephrased the relevant section in the discussion to: *“When MT polymerization rates are increased in cells with a fully functional SAC, this will lead to decreased cell motility and increased kinetochore-MT stability and thus hyper-stable kinetochore-MT interactions. In this setting, the SAC will still delay mitosis until all chromosomal abnormalities caused by Src inhibition are resolved. However, when the SAC is also inhibited, it can no longer resolve the hyperstable kinetochore-MT interactions caused by Src inhibition, thus further increasing the frequency of chromosome missegregation events (Fig. 6).”* We hope this clarifies this issue.

The cancer cell lines HT29 and SW620 which show high CIN are not affected by Src inhibition. Therefore, SKI606 does not target CIN or aneuploid cells, it only inhibits the proliferation of cells where Mad2 has been knockdown. Do the other drugs identified in the screen (AZD8055 or EPZ015666) target CIN cells?

>This is an important point and focus of our ongoing work. Given time constraints, and the fact that the first author moved to pursue her first postdoc, we decided to only follow up on our single strongest hit in this study, and solve on the underlying mechanism of the toxic interaction. We will do the same for AZD8055 and EPZ015666 in a separate forthcoming study that will also include the mechanism of action for these drugs.

To validate the specificity of Src inhibition, cells were treated with another inhibitor of Src kinase (SKI-1). Indeed, they show that this inhibitor also inhibits the proliferation of Mad2-kD cells. However the grow curves profiles of cells treated with these two drugs look very different (4G vs. 4H) pointing toward different mechanisms of action. An independent method should be used to validate that indeed that inhibition of Src kinase activity is responsible for the selective inhibition of proliferation of Mad2-kD cells.

>This is a very important point, which was also brought up by reviewer 2, and we copied our response to this issue hereunder: While this reviewer does not ask for genetic perturbation of Src, we fully agree this is an important point as many drugs (even when testing multiple) have off-target effects. We therefore engineered inducible shRNA constructs targeting Src and now show that genetic perturbation of Src phenocopies the effect of SKI-606. We added the following text to the results: *“However, as small molecule compounds can have (overlapping) off-target effects, we also wanted to confirm the synergy between Src inhibition and an impaired SAC at the genetic level. For this, we designed 3 inducible shRNA constructs for Src of which one (shRNA3) yielded a significant knockdown of Src protein levels in RPE1 cells (54% knockdown, Sup. Fig. 3N). Indeed, we found that RPE1 SrcKD cells were much more sensitive to the SAC inhibitor Reversine than wildtype RPE1 cells (Fig. 4I), particularly during the second half of the experiment (days 4-7), similar as observed for Mad2cKD cells treated with SKI606 (compare Fig. 4H to 4I). We therefore conclude that pharmaceutical as well as genetic inhibition of Src is selectively toxic to cells with an impaired spindle assembly checkpoint.”* We added panel I to Figure 4 and panel N to Sup. Figure 3 showing

the data and added some text to the discussion as well: *“Importantly, we validated the phenotype with another Src inhibitor and confirmed that the effect of the inhibitors is caused by Src inhibition, since genetic perturbation of Src by shRNA with SAC inhibition yields the same phenotype as inhibitor treatment with SAC inhibition. Of note, we only succeeded in reducing Src protein levels by approximately 2-fold using shRNA and failed to engineer Src knockout cell lines using CRISPR engineering (data not shown), which suggests that cells critically rely on some remaining Src kinase activity for their survival. Therefore, in order to phenocopy the selective targeting of CIN cells in vivo in future studies, it will be important to titrate drug concentrations well.”* Note that we also tried engineering Src KO lines using CRISPR engineering, a technique that works very efficiently in the lab. In this case, we tested 3 different guideRNAs in 6 independent transfection rounds which yielded no Src KO cell lines, strongly suggesting that Src loss is toxic. However, with this new data, we are convinced that the synergy between SAC inhibition and SKI-606 is indeed explained by Src inhibition.

Confusingly, Src inhibition does not affect the proliferation or cause a mitotic phenotype in RPE1 or MCF7 (Figures 4A and 4C), yet this drug affects cell motility and microtubule polymerization. Do these data suggest that altering microtubule dynamics do not affect mitosis?

>We fully understand the confusion. We believe that the mitotic abnormalities that Src inhibition provokes at $t0.5 \mu\text{M}$ are all repaired when the SAC is fully functional as it is in wildtype RPE1 cells. We clarified this point further by explicitly mentioning the increased MT polymerization rates together with the hyperstable MT-kinetochore interactions, which hopefully clarifies this point. The relevant text now reads: *“When MT polymerization rates are increased in cells with a fully functional SAC, this will lead to decreased cell motility and increased kinetochore-MT stability and thus hyper-stable kinetochore-MT interactions. In this setting, the SAC will still delay mitosis until all chromosomal abnormalities caused by Src inhibition are resolved. However, when the SAC is also inhibited, it can no longer resolve the hyperstable kinetochore-MT interactions caused by Src inhibition, thus further increasing the frequency of chromosome missegregation events (Fig. 6).”*

The graphs showing the effects of the drugs (1A-D, 3A-D) are very confusing. It is not clear what "log2 difference" reports. Does log2 difference equal to 1 mean that the area under the curve of control minus cells treated with the drug is 2? Do the authors mean log2 of the ratios? If this is the case a value of 1 means that the AUC has increased 2 fold upon drug treatment? Or decrease by 50%?

>We thank the reviewer for this comment. The reviewer interpreted the graphs correctly, but we agree that the labeling of the previous version of Figs 1 and 3 was far from intuitive. We therefore clarified the text in the Materials and Methods for the AUC description and changed the labeling of the axes in Figs 1 and 3 to be clearer. Furthermore, we transformed the log values into the outcomes of the log values and plotted those instead of the log-values. Altogether this should make the interpretations of Figs 1 and 3 much more intuitive.

The axis of several figures is mislabeled log (%Confluency). A log₁₀ of 100% would be 2.

>The reviewer is completely correct. The numbers at the y-axes are the outcomes of the log values (not the log values themselves), and the axes are plotted at the log-scale. We changed the labels of the y-axes to “Confluency (%)” to correct this.

1407 toxic only after 5-8 days. Is the reason for this result since it takes several days for the cells to become highly aneuploid? Does 1407 treatment affect the karyotype of cells?

>1407 only affects the karyotype of the cells when cells also have an impaired SAC. This is now much better explained in the discussion. We indeed believe that this is why it takes some time for 1407 to yield an effect: Src inhibition will lead to deregulated MT-kinetochore interactions that will in turn lead to an increased CIN rate and thus further growth inhibition.

The authors use the word killing throughout the manuscript. However, lower confluency could be due to inhibition of proliferation not due to increased cell death. The levels of cell death or apoptosis were not reported.

>This is absolutely true; we did not discriminate between cell death and cell cycle arrest or increased cell cycle timing. We replaced ‘killing’ for “prevented accumulation of” or “target” or something similar throughout the manuscript. Furthermore, we added a paragraph to the discussion further highlighting this: *“While we have shown in this study that Src inhibition decreases mitotic fidelity specifically in cells with an inhibited spindle checkpoint, we have not further investigated the downstream consequences of the resulting increased CIN rates. It is likely that the growth defects we observed following combined Src and SAC inhibition are caused by a combination of cell cycle arrest and cell death (for a recent review on this topic see e.g. (Chunduri & Storchová, 2019)), and that whether cells arrest or undergo apoptosis depends on the karyotypes that the aneuploid cells acquired following the CIN insult. As the consequences of CIN can be highly tissue-specific (Foijer et al, 2013, 2017), it will be important to further investigate the downstream consequences of altering MT dynamics in combination with spindle checkpoint inhibition in cell and organoid cultures and in animal models before translating these findings to the clinic.”*

In the third paragraph of the introduction, what does CIN (28) mean?

>Thanks for noting this. This was an improperly formatted reference at the wrong place. This is now corrected.

2nd Editorial Decision

03 January 2020

January 3, 2020

RE: Life Science Alliance Manuscript #LSA-2019-00499-TR

Dr. Floris Foijer
University Medical Center Groningen
European Institute for the Biology of Aging
Antonius Deusinglaan 1
Groningen 9713AV
Netherlands

Dear Dr. Foijer,

Thank you for submitting your revised manuscript entitled "Altering microtubule dynamics is synergistically toxic with spindle assembly checkpoint inhibition". I have now assessed the revisions performed and appreciate the introduced changes. I think that they address the reviewer concerns well. I would thus be happy to publish your paper in Life Science Alliance pending final revisions necessary to meet our formatting guidelines:

- Please provide the ms file in word docx format
- Please upload the suppl. Movies and include a short legend for these in the main manuscript docx file
- PRJEB33217 does not exist in ENA, please release

A. FINAL FILES:

-- Summary blurb (enter in submission system): A short text summarizing in a single sentence the study (max. 200 characters including spaces). This text is used in conjunction with the titles of papers, hence should be informative and complementary to the title. It should describe the context and significance of the findings for a general

readership; it should be written in the present tense and refer to the work in the third person. Author names should not be mentioned.

B. MANUSCRIPT ORGANIZATION AND FORMATTING:

Sincerely,

Andrea Leibfried, PhD
Executive Editor
Life Science Alliance
Meyershofstr. 1
69117 Heidelberg, Germany
t +49 6221 8891 502
e a.leibfried@life-science-alliance.org
www.life-science-alliance.org

3rd Editorial Decision

08 January 2020

January 8, 2020

RE: Life Science Alliance Manuscript #LSA-2019-00499-TRR

Dr. Floris Fojer
University Medical Center Groningen

European Institute for the Biology of Aging
Antonius Deusinglaan 1
Groningen 9713AV
Netherlands

Dear Dr. Fojjer,

Thank you for submitting your Research Article entitled "Altering microtubule dynamics is synergistically toxic with spindle assembly checkpoint inhibition". I appreciate the introduced changes and it is a pleasure to let you know that your manuscript is now accepted for publication in Life Science Alliance. Congratulations on this interesting work! Please note that I added callouts to the suppl. movies next to the callout to Figure 5F in the manuscript text.

DISTRIBUTION OF MATERIALS:

Again, congratulations on a very nice paper. I hope you found the review process to be constructive and are pleased with how the manuscript was handled editorially. We look forward to future exciting submissions from your lab.

Sincerely,
